# LARGE MULTIMODAL MODEL FOR REAL-WORLD RADIOLOGY REPORT GENERATION

## ABSTRACT

While automatic report generation has demonstrated promising results using deep learning-based methods, deploying these algorithms in real-world scenarios remains challenging. Compared to conventional report generation, real-world report generation requires model to follow the instruction from the radiologists and consider contextual information. Thus, this paper focuses on developing a practical report generation method that supports real-world clinical practice. To tackle the challenges posed by the limited availability of clinical data, we propose a GPT-based unified data generation pipeline designed to produce high-quality data. Consequently, we present a new benchmark dataset *MIMIC-R3G*, comprising five representative tasks pertinent to real-world medical report generation. We propose Domain-enhanced Multi-modal Model (*DeMMo*), where an additional medical domain vision encoder is incorporated into the general domain multi-modal LLM to enhance its ability on specific domains. This approach aims to harness the specialized capabilities of the medical domain vision encoder while leveraging the robustness and versatility of the general domain multi-modal LLM. Comprehensive experiments demonstrate that our approach attains competitive performance across all real-world tasks compared to existing interactive report generation frameworks and state-of-the-art encoder-decoder style report generation models.

## 1 INTRODUCTION

Radiology report generation is one of the straightforward yet essential task in computer-aided diagnosis (CAD) systems. It aims to automatically generate a text description of the patient's radiology images including professional medical diagnosis. Recent works can automatically generate radiology report accurately within seconds, which largely reduces the workload of professional radiologists in clinical routines (Jing et al., 2018; Chen et al., 2020; Liu et al., 2021; Wang et al., 2022a; Huang et al., 2023).

Most previous works treat radiology report generation as a captioning task, where a text decoder generate medical report based on extracted image features (Nicolson et al., 2023). In real clinical practice, however, the scenario and procedure might be more complex than a straightforward captioning task. Specifically, in real-world scenarios, the model is required to follow broader instructions of the radiologists and to consider different types of context information. For example, radiologists usually need to refer to the patient's X-ray images and reports from previous visits in order to write a more comprehensive report that includes progress or changes in the abnormalities. Also in many cases, patients are required to undergo some other medical examinations beside radiology screenings. All these kinds of extra information could affect how radiologists read the radiographs and write the final report for the patient. Therefore, this paper focuses on developing a practical report generation method that supports real-world clinical practice containing various interactions and external context information.

Large Language Models (LLMs) pose significant potential in performing real-world report generation tasks, owning to their capabilities in interaction, instruction following, medical domain knowledge and long sequence generation. However, to develop a LLM based real-world report generation system, there are two challenges to overcome. The first challenge is the scarcity of clinical data that comprises different scenarios, which could hinder the development of robust LLMs. Current med-

ical report generation datasets are predominantly obtained from hospital or clinical databases. The information available in these datasets is generally limited to medical images and associated structured reports (Johnson et al., 2019; Demner-Fushman et al., 2016), lacking supplementary information that might influence radiologist's reasoning in formulating a diagnosis. The second challenge involves fine-tuning LLMs, which typically possess billions of parameters to optimize, requiring considerable computational resources. Specifically for medical domain, the training process can be laborious, as it is vulnerable to overfitting, particularly when dealing with a relatively small amount of data.

To address the challenges in data scarcity, we examine the real-world clinical requirements and propose a new benchmark dataset, named *MIMIC-R3G* (Real-world Radiology Report Generation). *MIMIC-R3G* contains five representative tasks pertinent to the medical report generation context: report generation with no context, report revision, template-based report generation, report generation based on patient's previous visits, and report generation incorporating patient's other information including medical records and laboratory tests. Building on these tasks, we introduce a unified automatic data generation pipeline to generate instructions, context, and reports in accordance with the ground truth report and images, using specific system messages and ground truth reports as input to direct ChatGPT (OpenAI, 2022) for generation. To address the challenge of fine-tuning LLMs for real-world report generation, we propose Domain-enhanced Multimodal Model (*DeMMo*), where a medical domain vision encoder is incorporated into Flamingo (Alayrac et al., 2022), a pretrained general domain large multimodal model. *DeMMo* effectively enhances the domain-specific capabilities of the pretrained LLM while retaining its general medical domain knowledge. Comprehensive experiments on the *MIMIC-R3G* benchmark demonstrate that our method achieves comparable results on all real-world tasks, compared to existing interactive report generation framework and state-of-the-art encoder-decoder style report generation models.

In summary, the contributions of this paper are as follows:

- We present a new problem setting for real-world report generation that emulates clinical practices by incorporating various clinical interactions and contextual information.
- We propose the first real-world report generation benchmark dataset *MIMIC-R3G*, where a unified framework designed to automatically generate the requisite context data, leveraging the power of LLM.
- We develop *DeMMo* , a large multimodal model with domain-specific capability enhanced via incorporating a general domain Flamingo with an additional medical vision encoder.

The rest of the paper is organized in the order of problem stating, data, solution and results. Section 2 introduces the related works. Section 3 elaborates on the real-world radiology report generation problem setting. Section 4 introduces the unified data generation pipeline and the new benchmark datasets *MIMIC-R3G*. Section 5 introduces our large multimodal model *DeMMo*. Section 6 and 7 are the experiments and conclusions. Due to the page limit, more details on our work are introduced in the Appendix.

## 2 RELATED WORKS

In this section, we review the literature on medical report generation, specifically on recent deep learning based methods. Traditional methods use an encoder-decoder regime, where an encoder is used to extract image features, and a decoder is used to generate text from the features. The combination of CNN encoder and RNN decoder were utilized in earlier works (Jing et al., 2018; Xue et al., 2018; Wang et al., 2018; Hou et al., 2021). With the advent of Transformer architecture, researchers have explored the use of Transformer with specialized memory or attention mechanisms for report generation (Cornia et al., 2020; Chen et al., 2020; 2021; You et al., 2021). To further improve performance, many works incorporated pre-extracted pathology labels and domain-specific knowledge graphs as priors in the generation pipeline (Liu et al., 2021; Wang et al., 2022b; Huang et al., 2023; Li et al., 2023d). Some retrieval-based approaches have also gained prominence in recent years (Endo et al., 2021; Jeong et al., 2023). These methods predominantly employ contrastive learning techniques to retrieve probable texts from the training set as inference outcome. Building on existing approaches, several studies (Wu et al., 2022; Zhu et al., 2023) have also taken real-world clinical scenarios into account, but primarily focusing on the single task of incorporating reports from pre-

vious visits as a generation prior. We expand on this and propose a unified task formulation of real-world report generation. Recently, several works have been proposed to adapt Large Language Models (LLMs) to medical domain, which poses great potential for real-world report generation. In this study, we propose employing multimodal LLMs for report generation task. We discuss more related works about multimodal LLMs in Appendix A.

## 3 R3G: REAL-WORLD RADIOLOGY REPORT GENERATION PROBLEM

In contrast to conventional report generation models, Real-world Radiology Report Generation (R3G) poses two significant differences. Firstly, it necessitates the model to adhere to the user's requests and instructions. Secondly, in addition to the medical image itself, the model must possess the capability to comprehend and utilize external contextual information in order to produce a more precise report. As a results, we propose several representative sub-tasks that resembles these two requirements, all of which are essential features widely applicable in clinical practice. The instances drawn from these representative sub-tasks will be used to train and evaluate our proposed report generation model.

**Report generation.** This sub-task is the conventional report generation task without any additional instructions from radiologist or context information.

**Report revision.** Reports generated models may be sub-optimal in some cases, and and human professionals are still required to review and revise the output reports prior to submission. Therefore, it is desirable for the model to possess the capability of revising the report based on straightforward instructions to further alleviate the workload of the human professional.

**Template.** In real-world scenarios, clinics or hospitals may employ structured report templates. These templates may comprise a list of common abnormalities or regions, and the radiologist is required to fill in the corresponding findings or absence of abnormalities. In sum, we want the model to be capable of generating report following any form of input template.

**Previous Radiology Image and Report as Context.** In typical clinical practice, patients undergo multiple radiology screenings. It is essential for radiologists to write medical reports that not only focus on the current radiology image but also reference the patient's previous medical images and reports. This approach enables the production of a more informative report that can address the alterations in the disease progression compared to previous visits.

**Medical Records and Lab Tests as Context.** Patient's medical records, including medical condition history, along with medical exams like blood tests and pulmonary function tests, are vital for accurate diagnosis. Medical records and lab tests are all crucial context information for radiologists to write reports, so the model should also posses the ability generate reports based on them.

## 4 *MIMIC-R3G*: DATASET FOR REAL-WORLD REPORT GENERATION

### 4.1 TASK FORMULATION

We formulate the proposed real-world report generation tasks under a unified instruction-following paradigm, so we can fully utilize the instruction-following capabilities of a Large Language Model (LLM). Specifically, we format the proposed real-world report generation tasks into a unified single-round instruction-following example: $(V_i, I_i, C_i, R'_i)$, representing the $i$-th example in the dataset, where $V_i$ denotes a set of medical images; $I$ denotes the instruction from the user; $C_i$ refers to the context information provided to facilitate the report generation; and $R'_i$ refers to the ground truth report associated with the medical images $V_i$, instruction $I_i$, and context $C_i$ in the generated dataset. For all the sub-tasks, $V_i$ is directly utilized from the dataset.

### 4.2 DATA GENERATION

Existing large-scale report generation datasets, such as MIMIC-CXR (Johnson et al., 2019), are not tailored for real-world report generation as they lack user instructions $I_i$ and contextual information $C_i$ paired with corresponding responsive report $R'_i$. The manual collection of such instructional and contextual data is prohibitively costly and may raise privacy concerns. Hence, we propose to harness

the capabilities of ChatGPT and construct a unified pipeline to automatically generate diverse and relevant real-world clinical text data based on existing ground truth reports in conventional datasets.

The primary goal is to either design or generate instructions $I_i$ and context $C_i$, and also possibly modify the ground truth report $R_i$ from dataset into $R_i'$ according to different sub-tasks. To generate a dataset of a single real-world report generation task, the objection of our pipeline is $\{(V_i, R_i)\}_{i=1}^N \mapsto \{(V_i, I_i, C_i, R_i')\}_{i=1}^N$, where $N$ is the number of examples of an existing report generation dataset.

The medical image $V_i$ stays un-changed and directly comes from the original dataset. We devise different task-specific system messages to generate the required $I_i$, $C_i$, and $R_i'$ for distinct tasks. Using the ground truth report $R_i$ as input, along with in-context examples (omitted in examples) to guide the output format, the response can be filtered and parsed accordingly into the required data components. Next we will elaborate on how request from each sub-task is organized as an instruction-following example, and how the examples are produced for each sub-task. We show one data generation example, and please refer to Appendix B for examples of other tasks.

**Report generation.** For basic report generation task, the data sample follows $(V_i, I_i, C_i, R_i')$, where $V_i$ and $R_i' = R_i$ are directly utilized from report generation dataset. $I_i$ is a manually designed instruction, and $C_i$ is kept empty.

**Report revision.** For report revision task, $R_i' = R_i$ come from the report generation dataset, $I_i$ is the instruction of how to revise or correct the report, and $C_i$ is the report that the user wants the model to revise. To generate $I_i$ and $C_i$ for this task, we employ our proposed pipeline to produce a slightly modified report based on the input ground truth report, along with the instructions of how to revise the modified report into the correct ground truth report. We show an example of system message used and ChatGPT response of report revision task here.

---

**Example of Generating Report Revision Data**

**System Message:** You will make some minor diagnosis error when reading a chest x-ray radiograph. You will be given a correct chest x-ray report with a findings section, and you will be asked to rewrite the report with a few diagnostic errors. Then provide the instructions of how to correct your wrong diagnosis.

**Input Report:** PA and lateral views of the chest provided. The lungs are adequately aerated. There is a focal consolidation at the left lung base adjacent to the lateral hemidiaphragm. There is mild vascular engorgement. There is bilateral apical pleural thickening. The cardio mediastinal silhouette is remarkable for aortic arch calcifications. The heart is top normal in size.

- - - - - - - - - - - - - - - - - - - - - - - - - - - - - - - - - - - - - - - - - -

**Response:** *Report:* PA and lateral views of the chest provided. The lungs are adequately aerated. There is no focal consolidation. There is mild vascular engorgement. There is no bilateral apical pleural thickening. The cardio mediastinal silhouette is remarkable for aortic arch calcifications. The heart is mildly enlarged in size.

*Instructions:* Add focal consolidation. The patient has bilateral apical pleural thickening.

- - - - - - - - - - - - - - - - - - - - - - - - - - - - - - - - - - - - - - - - - -

$I_i = \langle$Text in Instructions section$\rangle$
$C_i = \langle$Text in Report section$\rangle$

---

**Template.** $I_i$ is a manually designed instruction, *e.g.*, *Fill in the template based on the give medical images*. $I_i$ and $R_i'$ are the empty template and the corresponding filled template. We leverage our pipeline to first generate a free-form medical report template, and a structured version of the input ground truth report will be produced based on the generated template.

**Previous Visit as Context.** $I_i$ is manually designed instruction telling the model to generate report based on both the medical images and report from last visit. $C_i$ can be the retrieved previous report of the same patient from the dataset, and $R_i' = R_i$ is the ground truth report. In the case when previous report is unavailable, we can select a random report from the dataset as the context $C_i$ and employ the proposed pipeline to generate a modified report $R_i'$ from both the current report and $C_i$ as a pseudo previous report. The modified report $R_i'$ should have the diagnosis unchanged compared to $R_i$ but with more descriptions on comparisons between two reports. It should be noted that $V_i$ in this task can include medical images of the patient from their previous visit as well.

**Medical Records and Lab Tests as Context.** Similarly, $R_i' = R_i$ comes from the original dataset, and $I_i$ is a manually designed instructions. $C_i$ here represents the additional medical conditions or medical examination results that the patient may posses. Since the ground-truth medical report exhibits a strong correlation with the external context information, our pipeline generates $C_i$ by

|        | No Context | Revision | Template | Previous Report | Medical Record | Total   |
|--------|------------|----------|----------|-----------------|----------------|---------|
| #Train | 140,781    | 116,596  | 104,407  | 62,576          | 51,008         | 475,368 |
| #Test  | 2,020      | 1,377    | 1,273    | 1,253           | 643            | 6,566   |

Table 1: Statistics of *MIMIC-R3G*

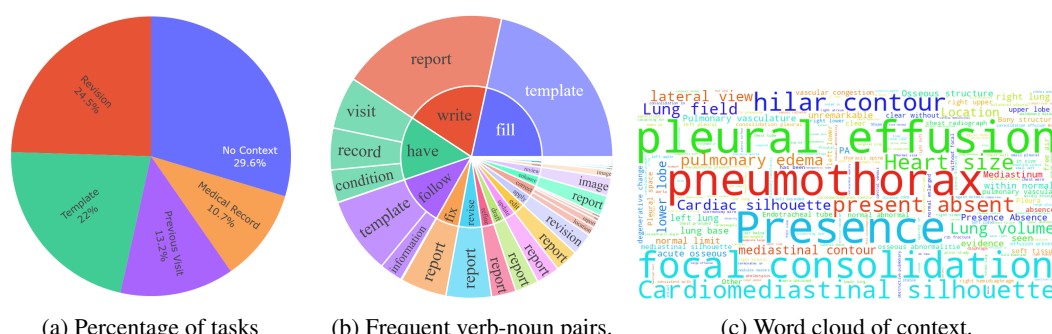

(a) Percentage of tasks     (b) Frequent verb-noun pairs.     (c) Word cloud of context.

Figure 1: Visualizations of *MIMIC-R3G* statistics. (a) shows the general distribution of data of different tasks. (b) shows the frequent verb-noun pairs appeared in instructions of our dataset, where inner circle is the root verb, and outer ring represents the paired nouns.

inferring the plausible medical conditions, medical examinations and exam results based on the ground truth medical report.

Since the ground truth report $R_i'$ is either identical to original report $R_i$ or rewritten by ChatGPT while preserving the medical diagnosis intact, our pipeline is able to produce data with very few factual errors. Furthermore, the generated data has undergone both manual and automatic data quality exam and control processes. Specifically, a medical professional validates the clinical correctness of a subset of the generated data. Unsatisfactory generated reports are further filtered by extracting and comparing the generated context with the ground truth report. We use CheXpert labeler (Irvin et al., 2019), an automatic tool to extract labels of common observation from radiology reports, to extract and compare the labels of $C_i$ and $R_i'$ to ensure that no information leakage is presented in the generated context.

## 4.3 DATASET STATISTICS AND ANALYSIS

Using our data generation pipeline, we generate a novel dataset based on a large report generation dataset MIMIC-CXR (Johnson et al., 2019), named *MIMIC-R3G*. Since MIMIC-CXR already contains patients' previous reports, we directly use the report from dataset as ground truth and retrieved previous report as context without generation. The statistics of original MIMIC-CXR data and generated data for all tasks is shown in table 1.

A subset of the generated dataset has undergone validation by a medical professional. Specifically, approximately twenty samples from each sub-task were randomly selected, and a doctor evaluated the quality of each sample to ensure that the generated content are suitable with its corresponding ground-truth report. In summary, our medical professionals have determined that no significant factual discrepancies exist between the content generated by GPT and the ground-truth reports across all samples. This validation affirms the capability of our pipeline to generate clinically correct content with no factual errors. However, in the case of relatively challenging sub-tasks, such as using medical records and lab tests as context, some of the context information generated by GPT may lack informativeness and provide limited assistance in report generation. Specifically, in 1 out of 22 samples, we observed a weak correlation between the generated medical condition context and the findings of the original report (e.g., generating a hemorrhagic condition in a case with PICC placement). In 2 out of 22 samples, the generated diagnosis might not accurately reflect the most common relationship between disease and image finding (e.g., generating a lung cancer condition in a pulmonary atelectasis case). Although this represents a limitation in our current data generation pipeline that requires future improvement, we believe that including samples with less informative

---

**No Context**

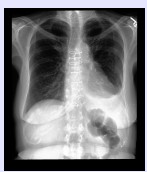

**Instruction:** Act as a radiologist and write a diagnostic radiology report for the patient based on their chest radiographs
**Report:** The patient is status post recent median sternotomy and coronary artery bypass surgery. Cardiomediastinal contours are stable in appearance compared to previous post-operative radiographs. Moderate left pleural effusion has slightly increased in size since the previous radiograph, and is associated with adjacent left basilar atelectasis. On the other hand, a small right pleural effusion has decreased in size and nearly resolved.

---

**Report Revision**

**Instruction:** Update the medical report of the given chest x-ray images with these changes: replace "significantly increased" with "slightly increased"
**Context:** Report: The patient ... Moderate left pleural effusion has *significantly increased* in size since the previous radiograph, and is associated with adjacent left basilar atelectasis ... still present.
**Report:** (same as no context)

---

**Medical Record as Context**

**Instruction:** Based on the chest x-ray images and patient's medical details, draft a detailed diagnostic medical report
**Context:** Medical conditions of the patient: Echocardiogram shows decreased cardiac output or decreased ejection fraction, pulmonary function tests show decreased lung capacity or decreased oxygen saturation
**Report:** (same as no context)

---

**Template**

**Instructions:** Please act as a radiologist and write a diagnostic radiology report for the patient based on their chest radiographs, the format should follow the template
**Context:** Template:
Cardiomediastinal contours: [stable/unstable]
...
bony structures: [normal/abnormal]
**Report:**
Cardiomediastinal contours: stable.
...
Bony structures: not mentioned.

---

**Previous Visit as Context**

**Instruction:** Please write a diagnostic radiology report for the patient based on their chest radiographs considering the report from last visit
**Context:** Medical report from last visit: Following removal of left-sided chest tube, there is a probable residual tiny left apical pneumothorax. Otherwise, no short interval change in the appearance of the chest since the recent study performed earlier the same date.
**Report:** (same as no context)

Table 2: Examples of *MIMIC-R3G* generated using the same report on different tasks

context information is still pertinent to real-world report generation scenarios, where context information may not always perfectly align with radiology findings.

## 5 METHODOLOGY

For real-world report generation, our objective is to train a model that given the image-text input $x = (V, I, C)$ generate output text $y = R'$, therefore the generation process can be formalized as $p_\theta(y \mid x)$ where $\theta$ represents the model parameters to be optimized. Our model is built upon Flamingo (Alayrac et al., 2022) due to its training efficiency and good performance. To further enhance the domain specific capability of the general domain flamingo model, *DeMMo* inserts an additional domain specific medical encoder to the perceiver resampler of flamingo. A parameter efficient prompt tuning method is adopted to fine-tune the model for medical domain while preserving its generalization ability.

### 5.1 FLAMINGO AS MODEL BASIS

Flamingo is a family of vision language model that is capable of generating language conditioned on interleaved text and image sequences. By connecting visual encoder and LLM with a perceiver resampler, Flamingo models the likelihood of text output $y$ conditioned on interleaved image and text input as a next-token prediction task: $p_\theta(y \mid x) = \prod_{l=1}^{L} p(y_l \mid y_{<l}, x_{<l})$, where $y_{<l}$ and $x_{<l}$ are the sets of text and image tokens preceding $y_l$, the $l$-th input text token.

The general domain visual encoder of Flamingo exhibit greater diversity and generalization ability, but cannot fully capture the detailed visual feature in medical domain. Consequently, a domain specific encoder is required to capture the nuances and specific characteristics of medical images. In this paper, we employ BioViL (Boecking et al., 2022) as our medical vision encoder. To capitalize on the robust generalizability and expedite convergence, the original pretrained general domain visual encoder in Flamingo is still preserved in conjunction with the newly introduced medical encoder.

## 5.2 MODEL ARCHITECTURE AND FINE-TUNING APPROACH

We propose Domain enhanced Mulimodal Model (*DeMMo*), which incorporates a domain specific visual encoder into the general domain Flamingo.

As shown in figure 2, given a set of images $V_i$ that contains $k$ images, the original Flamingo vision encoder outputs $n \times n$ grid features $\boldsymbol{X}_f \in \mathbb{R}^{k \times n \times n \times d_f}$, and the medical vision encoder outputs an $m \times m$ grid features $\boldsymbol{X}_m \in \mathbb{R}^{k \times m \times m \times d_m}$, where $d_f$ and $d_m$ are feature dimensions of Flamingo vision encoder and medical vision encoder, respectively. After applying a projection $\boldsymbol{W} \in \mathbb{R}^{d_m \times d_f}$ to $\boldsymbol{X}_m$ followed by flattening both grid features, we get $\boldsymbol{X}_f \in \mathbb{R}^{kn^2 \times d_f}$ and $\boldsymbol{X}_m \in \mathbb{R}^{km^2 \times d_f}$. We adopt the idea of LLaMA-Adapter (Zhang et al., 2023) to insert a learnable adaption prompt $\boldsymbol{P}_l \in \mathbb{R}^{m^2 \times d_f}$ into the perceiver resampler independently for each layer $l$. Each flattened feature from medical vision encoder is then added element-wise to $\boldsymbol{P}_l$ to form the medical visual feature prepared for attention. Similar to vanilla Flamingo, a predefined number of latent queries are cross-attended to the concatenation of queries and visual features. Formally, denote $t$ as the number of latent queries. At layer $l$, $\boldsymbol{Q}_l \in \mathbb{R}^{t \times d_f}$ is the latent queries, and $\boldsymbol{V}_l = \boldsymbol{K}_l \in \mathbb{R}^{\left(km^2 + kn^2 + t\right) \times d_f}$ is the concatenation of medical visual features, original visual features from Flamingo vision encoder, and the latent queries. Then, the similarity scores are computed as $S_l = \left(\boldsymbol{Q}_l \boldsymbol{W}_l^Q\right) \left(\boldsymbol{K}_l \boldsymbol{W}_l^K\right)^\top / \sqrt{d_h} \in \mathbb{R}^{t \times \left(km^2 + kn^2 + t\right)}$, where $\boldsymbol{W}_l^Q, \boldsymbol{W}_l^K \in \mathbb{R}^{d_f \times d_h}$ are query and key projections respectively at layer $l$, and $d_h$ represents the hidden feature dimension.

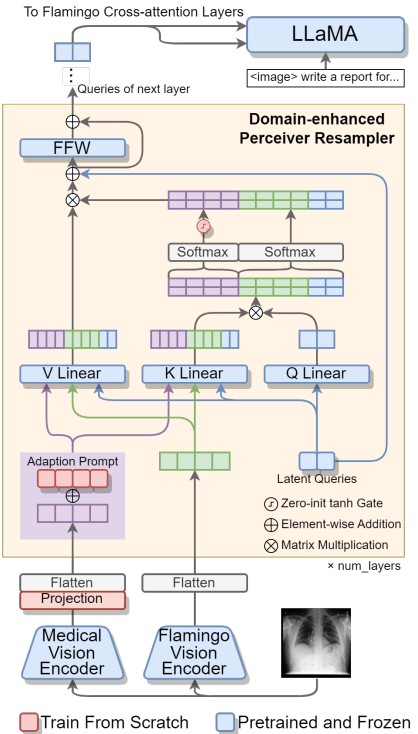

Figure 2: Architecture of *DeMMo*

Since the linear projection and the adaption prompt is randomly initialized, training with newly introduced medical visual features might introduce instability on top of the pretrained weights. Therefore, we follow Zhang et al. (2023) to independently apply softmax on the similarity score corresponding to the medical visual features. A zero-initialized $\tanh$ gate is utilized on the corresponding output attention score, ensuring a gradual increase of influence due to medical visual features. The rest are same as vanilla Flamingo model, where the attended queries pass through another feed-forward network before next layer, and the last perceiver layer output is inserted into Flamingo Cross-attention layers. We only tune the medical vision encoder projection, adaption prompts, zero-initialized gates, along with the cross-attention layer in LLaMA.

## 6 EXPERIMENTS

### 6.1 DATASET AND METRICS

Following Chen et al. (2020; 2021); Wang et al. (2022a); Nicolson et al. (2023), we use the frontal view images in *MIMIC-R3G* and focus on generating the findings. This results in 140,781 reports and 156,969 images in training set and 2020 reports and 2274 images in test set. We adopt natural language generation (NLG) metrics that measures text similarity between generated and ground truth report, including BLEU (B@n) (Papineni et al., 2002), METEOR (M) (Banerjee & Lavie, 2005), and ROUGE-L (R-L) (Lin, 2004). Following previous works, we also utilize CheXpert, an automatic labeler tool to extract observation labels from chest X-ray reports, to evaluate clinical efficacy (CE) in terms of micro-averaged label precision (P), recall (R), and F1-score (F1).

| Task | Method | B@1 | B@2 | B@3 | B@4 | M | R-L | P | R | F1 |
|------|--------|-----|-----|-----|-----|---|-----|---|---|-----|
| No Context | ChatCAD+ | 0.201 | 0.106 | 0.060 | 0.037 | 0.268 | 0.162 | 0.336 | 0.625 | 0.437 |
| | ChatCAD+* | 0.129 | 0.067 | 0.038 | 0.024 | 0.244 | 0.125 | 0.337 | **0.668** | 0.448 |
| | Enc-Dec | 0.314 | 0.176 | 0.112 | 0.076 | 0.121 | **0.260** | 0.399 | 0.110 | 0.172 |
| | Ours | **0.370** | **0.227** | **0.147** | **0.104** | **0.294** | 0.243 | **0.502** | 0.439 | **0.469** |
| Revision | ChatCAD+ | 0.499 | 0.442 | 0.401 | 0.367 | 0.700 | 0.557 | 0.674 | 0.866 | 0.758 |
| | ChatCAD+* | 0.377 | 0.338 | 0.308 | 0.282 | 0.671 | 0.489 | 0.660 | 0.902 | 0.762 |
| | Enc-Dec | 0.276 | 0.167 | 0.110 | 0.076 | 0.127 | 0.245 | 0.412 | 0.280 | 0.333 |
| | Ours | **0.898** | **0.874** | **0.853** | **0.835** | **0.903** | **0.900** | **0.947** | **0.920** | **0.933** |
| Template | ChatCAD+ | 0.268 | 0.217 | 0.181 | 0.152 | 0.450 | 0.284 | 0.492 | 0.733 | 0.588 |
| | ChatCAD+* | 0.246 | 0.199 | 0.165 | 0.138 | 0.439 | 0.264 | 0.492 | **0.762** | 0.598 |
| | Enc-Dec | 0.151 | 0.073 | 0.041 | 0.026 | 0.139 | 0.180 | 0.408 | 0.203 | 0.279 |
| | Ours | **0.667** | **0.592** | **0.534** | **0.484** | **0.626** | **0.601** | **0.758** | 0.728 | **0.742** |
| Previous Report | ChatCAD+ | 0.231 | 0.131 | 0.080 | 0.050 | **0.304** | 0.179 | 0.415 | 0.631 | **0.501** |
| | ChatCAD+* | 0.221 | 0.125 | 0.077 | 0.051 | 0.297 | 0.176 | 0.405 | **0.632** | 0.494 |
| | Enc-Dec | 0.272 | 0.165 | 0.108 | 0.076 | 0.123 | 0.246 | 0.412 | 0.262 | 0.320 |
| | Ours | **0.344** | **0.214** | **0.141** | **0.101** | 0.297 | **0.253** | **0.512** | 0.419 | 0.461 |
| Medical Record | ChatCAD+ | 0.135 | 0.063 | 0.032 | 0.019 | 0.206 | 0.108 | 0.330 | 0.433 | 0.375 |
| | ChatCAD+* | 0.110 | 0.055 | 0.029 | 0.017 | 0.221 | 0.100 | 0.310 | **0.545** | 0.395 |
| | Enc-Dec | 0.206 | 0.116 | 0.070 | 0.045 | 0.103 | 0.210 | 0.290 | 0.207 | 0.242 |
| | Ours | **0.343** | **0.208** | **0.133** | **0.092** | **0.287** | **0.243** | **0.457** | 0.378 | **0.414** |

Table 3: Comparison of our model with baselines on the test sets of our real-world report generation dataset. ChatCAD+* refers to the ChatCAD+ model using our model trained on no-context data (i.e. original MIMIC-CXR data) as report generator. Enc-Dec refers to the CvT-212DistilGPT2 encoder-decoder model.

## 6.2 BASELINES

**ChatCAD+** ChatCAD+ (Zhao et al., 2023) is an interactive report generation framework that connects medical image disease classifier and report generator with ChatGPT and online knowledge databases.

**ChatCAD+ with replaced report generator** ChatCAD+ utilize an R2Gen model trained on MIMIC-CXR dataset as the report generator backbone. We replace this report generator by our proposed model trained solely on MIMIC-CXR data as well. As this baseline employs the same report generator as *DeMMo*, it offers a more equitable comparison to illustrate the effectiveness of training an end-to-end multi-modal LLM using task-specific data generation, rather than linking it to an LLM via text prompts.

**CvT-212DistilGPT2** We adapted CvT-212DistilGPT2 (Nicolson et al., 2023), a conventional report generation model to consider context and instructions. We then trained it on *MIMIC-R3G* to evaluate its real-world report generation performance.

## 6.3 MAIN RESULT

The models are trained on *MIMIC-R3G* for 8 epochs with 4 batch size in all experiments. We use an ADAMW optimizer with $\beta_1 = 0.9, \beta_2 = 0.999$ and weight decay of 0.01. The learning rate is set to 1e-4 with a 1000-step warm-up and a cosine decay schedule. More detailed implementation of the experiments are elaborated in Appendix D.1.

As shown in Table 3, we compare *DeMMo* with aforementioned baselines on each of the *MIMIC-R3G* sub-tasks, respectively. Note that the test splits of each sub-task are not identical and hence the performance is not comparable across difference tasks. In terms of most NLG metrics, *DeMMo* outperforms the baselines by a large margin on all tasks. In terms of CE metrics, *DeMMo* achieves the highest F1 score on 4 sub-tasks. While generally achieve low performance in terms of NLG metrics, methods using ChatCAD+ achieves the highest CE metrics on 'Previous Report' task. This is because ChatCAD+ employs an extra pretrained image disease classifier trained with additional annotations that gives more accurate prior information to the ChatGPT.

| Method | B@1 | B@2 | B@3 | B@4 | M | P | R | F1 |
|---|---|---|---|---|---|---|---|---|
| R2Gen (Chen et al., 2020) | 0.353 | 0.218 | 0.145 | 0.103 | 0.142 | 0.333 | 0.273 | 0.276 |
| CMN (Chen et al., 2021) | 0.353 | 0.218 | 0.148 | 0.106 | 0.142 | 0.334 | 0.275 | 0.278 |
| XPRONET (Wang et al., 2022a) | 0.344 | 0.215 | 0.146 | 0.105 | 0.138 | - | - | - |
| CvT-212DistilGPT2 (Nicolson et al., 2023) | **0.395** | **0.249** | **0.172** | **0.127** | 0.155 | 0.365 | 0.418 | 0.390 |
| *DeMMo* (Ours) | 0.387 | 0.236 | 0.152 | 0.107 | **0.302** | **0.486** | **0.445** | **0.465** |

Table 4: Comparison of *DeMMo* with conventional report generation methods. The highest and the second highest performance are highlighted in bold and underline respectively.

| Task | Metrics | w/o Medical Encoder | w/o General Encoder | Independent Resampler | *DeMMo* |
|---|---|---|---|---|---|
| Previous Report | BLEU@1 | 0.343 | **0.346** | **0.346** | 0.344 |
| | Precision | 0.433 | 0.511 | 0.497 | **0.512** |
| | Recall | 0.336 | 0.409 | 0.388 | **0.419** |
| | F1 Score | 0.378 | 0.454 | 0.436 | **0.461** |
| Medical Record | BLEU@1 | 0.318 | 0.339 | **0.377** | 0.343 |
| | Precision | 0.382 | 0.449 | 0.436 | **0.457** |
| | Recall | 0.295 | 0.338 | 0.304 | **0.378** |
| | F1 score | 0.333 | 0.393 | 0.358 | **0.414** |

Table 5: Ablation studies on the performance comparison of different components in *DeMMo*, including medical encoder, general Flamingo encoder, and unified resampler.

## 6.4 CONVENTIONAL REPORT GENERATION

To show the efficacy of our model architecture design, we also evaluated the performance of *DeMMo* on conventional report generation task. Specifically, we train *DeMMo* using the original MIMIC-CXR dataset to compare with other conventional report generation models under the same setting. For a fair comparison, only generation methods that do not use extra medical dataset, knowledge graphs, or disease label or image classifier are compared. The performance of the comparison methods are directly cited from papers. As shown in Table 4, our methods significantly outperform existing conventional report generation methods in terms of CE metrics and a comparable performance in terms of NLG metrics.

## 6.5 ABLATION STUDY

We conduct ablation experiments to compare the performance of three other model designs. Table 5 reports the performance comparison of two more difficult sub-tasks, please refer to Appendix D.2 for the complete results. *DeMMo* outperforms the vanilla Flamingo without using medical vision encoders, showing the importance of adopting a medical vision encoder to enhance the domain-specific ability. The second baseline does not preserve the original Flamingo visual encoder like *DeMMo*, instead it directly replaces it with a medical vision encoder. The comparison results verify that preserving the original visual encoder can retain its general domain knowledge and hence help the performance. Instead of fine-tuning the original resampler with additional prompts, the third baseline trains an independent perceiver resampler for the input from the medical visual encoder. Compared to this baseline, *DeMMo* achieves comparable BLEU score and significantly higher CE performance. This demonstrates that employing a unified resampler for both the medical encoder and the general encoder ensures improved interactions between the two sets of features and enhances performance.

## 7 CONCLUSIONS

In this paper, we propose a highly interactive real-world radiology report generation problem setting (R3G). R3G requires models to be highly interactive, to follow instructions and consider various context information. A new benchmark dataset for the real-world report generation is built with a unified data generation pipeline. A novel Domain-enhanced Multi-Modal (*DeMMo*) model is proposed to enhance the medical domain specific ability of conventional LLM. Experiments demonstrate that *DeMMo* attains competitive performance across all real-world tasks.

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

## A    RELATED WORKS

**Large Language Models**   With the strong ability in natural language processing and generation, Large Language Models (LLMs) have shown significant potentials in performing real-world report generation tasks. State-of-the-art LLMs (Brown et al., 2020; Touvron et al., 2023; Chowdhery et al., 2022) are highly interactive and capable of following instructions for various language tasks (Ouyang et al., 2022), making it poses high potential in dealing with real-world clinical scenarios. Furthermore, the extensive volume of training data equips LLMs with the capacity to internalize domain-specific knowledge and exhibit reasoning capabilities within the medical field. Without fine-tuning on specific medical dataset, ChatGPT (OpenAI, 2022) is tested to pass the US Medical Licensing Exams (USMLE) (Kung et al., 2023) showing its promising ability to reason and process language in the medical domain. Finally, LLMs, in contrast to traditional language models, demonstrate proficiency in generating more extensive and complex text sequences, making them well-suited for medical report generation tasks.

**Multimodal LLMs**   With the remarkable success of LLMs, researchers started to explore the possibility to integrate visual modality into LLMs for various visual-language tasks. Early works such as BLIP-2 (Li et al., 2023c) leveraged a query Transformer to connect visual features to LLM. Flamingo (Alayrac et al., 2022) introduced extra trainable layers within LLM in addition to a Transformer to bridge visual and language modalities. KOSMOS-2 (Peng et al., 2023) incorporated location tokens as input to facilitate visual grounding capabilities for Language Learning Models. PaLM-E (Driess et al., 2023) enables multimodal capabilities of LLM by introducing visual encoding to the continuous space of LLM. LLaVa (Liu et al., 2023) and MIMIC-IT (Li et al., 2023a) leveraged GPT/ChatGPT to build visual instruction tuning datasets and developed multimodal LLMs as general instruction-follow visual agents. Following their ideas, we construct a real-world report generation dataset by building a unified data generation pipeline leveraging ChatGPT.

**Medical LLMs**   Numerous works have applied LLM within the medical domain through fintuning a general domain LLM. Med-PaLM (Singhal et al., 2022) and Med-PaLM 2 (Singhal et al., 2023) are medical domain-specific language models developed through instruction fine-tuning based on general domain LLMs. Med-PaLM M (Tu et al., 2023) further fine-tunes PaLM-E to the medical domain using multimodal medical data for medical vision-language tasks. LLaVa-Med Li et al. (2023b) and Med-Flamingo (Moor et al., 2023) similarly fine-tune their general domain base models using domain-specific data to enhance medical question-answering and conversational capabilities.

Different from fine-tuning LLMs for medical domain, ChatCAD (Wang et al., 2023) and ChatCAD+ Zhao et al. (2023) interact with users by connecting medical domain models with ChatGPT via language prompts. We observe that this framework is capable of doing all of our proposed real-world report generation tasks without training using extra task-specific data.

## B    *MIMIC-R3G* GENERATION EXAMPLES

---

**Example of Generating Template Data**

**System Message:** You are a clinical assistant helping radiologists write detailed and well-formatted chest X-ray reports. Without referencing the given report, you should first write a detailed chest X-ray report template with the findings section have different sections considering different pathological observations. Then, you should fill the template you have based on the given chest X-ray report.
**Input Report:** The heart is normal in size. A density paralleling the right border of the heart is consistent with a gastric pull-up, better seen on prior CT from ___. There is no focal consolidation, pleural effusion or pneumothorax.

- - - - - - - - - - - - - - - - - - - - - - - - - - - - - - - - - - - - - - - - - - - - - - - - - - - - - - - -

**Response:** *Template:*
Cardiac silhouette:
Pulmonary vasculature:
Lung parenchyma:
Pleura:
Mediastinum:
Diaphragm:
Bony thorax:
Soft tissues:

*Filled Template:*
Cardiac silhouette: The heart is normal in size.
Pulmonary vasculature: Not mentioned.
Lung parenchyma: No focal consolidation.

---

Pleura: No pleural effusion or pneumothorax.
Mediastinum: A density paralleling the right border of the heart is consistent with a gastric pull-up, better seen on prior CT from ___.
Diaphragm: Not mentioned.
Bony thorax: Not mentioned.
Soft tissues: Not mentioned.

- - - - - - - - - - - - - - - - - - - - - - - - - - - - - - - - - - - - - - - - - -

$I_i = \langle$Manually designed instructions$\rangle$
$C_i = \langle$Text in Template section$\rangle$
$R'_i = \langle$Text in Filled Template section$\rangle$

---

**Example of Generating Previous Visit Data**

**System Message:** You are a clinical assistant that helps manage the chest x-ray diagnosis reports of a patient. You will be given two chest x-ray reports in chronological order, and you are asked to rewrite the second report with more focus on comparison with the previous report.
**Input Report:** First report: The heart is normal in size. A density paralleling the right border of the heart is consistent with a gastric pull-up, better seen on prior CT from ___. There is no focal consolidation, pleural effusion or pneumothorax. Second report: PA and lateral views of the chest provided. Postsurgical changes related to a gastric pull-through surgery account for opacity at the right medial lung base. Overall appearance is unchanged. There is mild blunting of the right CP angle likely representing Mild pleural thickening. No focal consolidation, effusion or pneumothorax is seen. Heart size appears normal. Hilar and mediastinal configuration is stable. Bony structures are intact.

- - - - - - - - - - - - - - - - - - - - - - - - - - - - - - - - - - - - - - - - - -

**Response:** PA and lateral views of the chest provided. Comparison is made with the prior chest x-ray. The postsurgical changes related to a gastric pull-up are again noted, with the opacity at the right medial lung base remaining unchanged. The density paralleling the right heart border is consistent with prior gastric pull-up, as previously described. Heart size appears normal, similar to the previous report. There is mild blunting of the right CP angle, likely representing mild pleural thickening, which is a new finding compared to the previous report. No focal consolidation, effusion, or pneumothorax is seen, in line with the prior study. Hilar and mediastinal configuration remain stable, and bony structures are intact.

- - - - - - - - - - - - - - - - - - - - - - - - - - - - - - - - - - - - - - - - - -

$I_i = \langle$Manually designed instructions$\rangle$
$C_i = \langle$First input report or retrieved from dataset$\rangle$
$R'_i = \langle$Response or ground truth report from dataset$\rangle$

---

**Example of Generating Medical Record Data**

**System Message:** You are a clinical assistant that helps radiologists retrieve information of patient's other medical records and examination results. You will be given a chest x-ray diagnosis report, and you need to infer the patient's possible medical conditions or history and other medical examination the patient should have done. You need to first give several examples of their possible medical conditions based on the report diagnosis, then list a few the medical examinations that the patient should have done, and finally give a few examples of the patient's medical examination results that may lead to the diagnosis in the given report. List none if the report indicates no acute cardiopulmonary disease. Please make sure the inferred possible medical condition does not include any information described in the report findings or impression. Also make sure the examples of exam results are consistent with the findings described in the given report.
**Input Report:** The heart is normal in size. A density paralleling the right border of the heart is consistent with a gastric pull-up, better seen on prior CT from ___. There is no focal consolidation, pleural effusion or pneumothorax.

- - - - - - - - - - - - - - - - - - - - - - - - - - - - - - - - - - - - - - - - - -

**Response:** *Possible Medical Conditions:* Postoperative changes related to esophagectomy and gastric pull-up, Delayed gastric emptying. Possible Medical Examinations: Gastrointestinal motility study, Abdominal ultrasound Esophagogastroduodenoscopy (EGD), Pulmonary function tests.
*Examples of Medical Examination Results:* Gastrointestinal motility study: Delayed gastric emptying with reduced motility in the gastric conduit. Abdominal ultrasound: Gastric conduit in place without signs of obstruction or inflammation. Esophagogastroduodenoscopy (EGD): Gastric conduit appears patent without signs of inflammation, stricture, or obstruction. Pulmonary function tests: Normal lung function without significant restrictive or obstructive patterns.

- - - - - - - - - - - - - - - - - - - - - - - - - - - - - - - - - - - - - - - - - -

$I_i = \langle$Manually designed instructions$\rangle$
$C_i = \langle$Sampled phrases from Possible Medical Conditions and Examples of MedicalExamination Results sections$\rangle$
$R'_i = \langle$Ground truth report from dataset$\rangle$

---

## C  DETAILS OF MODEL ARCHITECTURE

As shown in figure 2 and section 5.2, our approach introduce the medical visual features into the existing pretrained attention in the perceiver resampler of Flamingo using adaption prompts at each layer. After obtaining the similarity scores, to ensure that no instability will be introduced when initializing the model with medical feature introduced, we follow Zhang et al. (2023) to apply softmax independent on two splits of the similarity score matrix, one on the scores corresponding to the Flamingo visual features and latent queries, and the other one on the scores corresponding to the newly introduced medical visual features. Specifically, $S_l$ could be separated into: $S_l = \begin{bmatrix} S_l^m; S_l^f; S_l^q \end{bmatrix}$ where $S_l^m \in \mathbb{R}^{t \times km^2}$, $S_l^f \in \mathbb{R}^{t \times kn^2}$, $S_l^q \in \mathbb{R}^{t \times t}$ represent similarity scores of the queries with respect to medical features, Flamingo vision encoder features, and the latent queries, respectively. We then apply a $\mathtt{tanh}$ gate controlled by a zero-initialized trainable parameter

$g_l$. The resulting attention score at layer $l$ is:

$$\text{Attn}_l = \left[ \tanh(g_l) \cdot \text{Softmax}\left(S_l^m\right); \text{Softmax}\left(\left[S_l^f; S_l^q\right]\right) \right]$$

In this way, when the model is initialized, medical visual features will have zero effect, and the forward process is equivalent to the forward process of a pretrained vanilla Flamingo. As the training advances, the gate parameter $g_l$ will be updated to gradually introduce the influence from medical visual features.

## D   MORE ON EXPERIMENTS

| Task | Method | B@1 | B@2 | B@3 | B@4 | M | R-L | P | R | F1 |
|---|---|---|---|---|---|---|---|---|---|---|
| No Context | Flamingo | 0.340 | 0.206 | 0.131 | 0.091 | 0.278 | 0.231 | 0.418 | 0.365 | 0.390 |
| | +Medical Encoder | 0.366 | 0.225 | 0.146 | 0.104 | **0.294** | **0.243** | 0.498 | 0.429 | 0.461 |
| | +Indep. Attn. | 0.359 | 0.220 | 0.143 | 0.102 | 0.284 | **0.243** | 0.484 | **0.441** | 0.461 |
| | *DeMMo* | **0.370** | **0.227** | **0.147** | **0.104** | **0.294** | **0.243** | **0.502** | 0.439 | **0.469** |
| Revision | Flamingo | 0.866 | 0.836 | 0.811 | 0.789 | 0.865 | 0.859 | 0.906 | 0.877 | 0.892 |
| | +Medical Encoder | 0.871 | 0.843 | 0.820 | 0.799 | 0.874 | 0.871 | 0.919 | 0.882 | 0.900 |
| | +Indep. Res. | 0.894 | 0.874 | 0.857 | 0.841 | 0.901 | 0.907 | 0.937 | 0.915 | 0.926 |
| | *DeMMo* | **0.898** | **0.874** | **0.853** | **0.835** | **0.903** | **0.900** | **0.947** | **0.920** | **0.933** |
| Template | Flamingo | 0.664 | **0.597** | **0.544** | **0.498** | **0.630** | 0.607 | **0.763** | 0.712 | 0.737 |
| | +Medical Encoder | 0.652 | 0.583 | 0.529 | 0.482 | 0.622 | **0.610** | 0.747 | 0.696 | 0.720 |
| | +Indep. Res. | 0.613 | 0.554 | 0.507 | 0.466 | 0.595 | 0.584 | 0.744 | 0.671 | 0.706 |
| | *DeMMo* | **0.667** | 0.592 | 0.534 | 0.484 | 0.626 | 0.601 | 0.758 | **0.728** | **0.742** |
| Previous Report | Flamingo | 0.343 | 0.207 | 0.133 | 0.094 | 0.281 | 0.234 | 0.433 | 0.336 | 0.378 |
| | +Medical Encoder | **0.346** | **0.214** | 0.139 | 0.098 | 0.289 | 0.244 | 0.511 | 0.409 | 0.454 |
| | +Indep. Res. | 0.346 | 0.216 | 0.143 | 0.103 | 0.286 | 0.250 | 0.497 | 0.388 | 0.436 |
| | *DeMMo* | 0.344 | **0.214** | **0.141** | **0.101** | 0.297 | **0.253** | **0.512** | **0.419** | **0.461** |
| Medical Record | Flamingo | 0.318 | 0.189 | 0.117 | 0.079 | 0.271 | 0.226 | 0.382 | 0.295 | 0.333 |
| | +Medical Encoder | 0.339 | 0.210 | 0.136 | 0.092 | 0.291 | 0.248 | 0.449 | 0.338 | 0.393 |
| | +Indep. Res. | **0.377** | **0.236** | **0.158** | **0.113** | **0.299** | **0.252** | 0.436 | 0.304 | 0.358 |
| | *DeMMo* | 0.343 | 0.208 | 0.133 | 0.092 | 0.287 | 0.243 | **0.457** | **0.378** | **0.414** |

Table 6: Results of ablation studies on different model architectures. Flamingo is the vanilla Flamingo model trained on *MIMIC-R3G* dataset. +Medical Encoder refers to the model that directly replace the vision encoder in flamingo by the medical vision encoder. +Indep. Res. refers to the architecture that simply add another resampler that process input latent queries and medical visual features in each layer.

### D.1   DATASETS AND IMPLEMENTATION DETAILS

The proposed generated benchmark datasets are built upon the ground-truth report in MIMIC-CXR, which is the largest widely used report generation dataset. It consists of chest X-ray radiographs and reports of 227,835 studies from 64,588 patients, with a total of 227,835 reports and 377,110 x-ray images. The official training and test splits of MIMIC-CXR includes 386,960 images and 222,758 reports in training set and 5159 images and 3269 reports in test set.

We adopt OpenFlamingo (Awadalla et al., 2023), which is an opensource implementation of the Flamingo architecture. We use BioViL (Boecking et al., 2022) as our medical vision encoder. The BioViL medical vision encoder outputs a $15 \times 15$ grid of features with feature dimension 2048, which is then flattened and projected into 225 1024-dimensional vectors, which is same as the feature dimension of original CLIP ViT-L/14 encoder in Flamingo. The length of adaption prompt in perceiver sampler is same as the number of visual features from medical vision encoder output, which is 225. We maintain other model design parameters, *e.g.*, hidden dimension and number of attention heads, consistent with the OpenFlamingo implementation. For each data sample, we randomly sample two frontal view chest x-ray images associated with the study, or add a dummy zero-valued image if there is only one available frontal view image. We train the model on *MIMIC-R3G* data for 8 epochs with 4 batch size in all experiments. We use an ADAMW optimizer with $\beta_1 = 0.9$, $\beta_2 = 0.999$ and weight decay of 0.01 and set the learning rate 1e-4 with a 1000-step warm-up and a cosine decay schedule. Beam search with beam size of 4 is used for report generation.

## D.2 ABLATION STUDIES

Table 6 reports the performance comparison result on all sub-tasks in *MIMIC-R3G*. Although *DeMMo* underperforms other designs on some NLG metrics, it achieves highest F1-scores on all tasks, proving the clinical efficiency of our design. This also shows that NLG metrics may not accurately reflect the performance of language generation model in the context of medical domain.

## D.3 USE CASES

In this section we present inference results of out model on all proposed report generation tasks. Input text prompt to the model is the concatenation of context and instruction in arbitrary order.

---

**Inference Examples of No Context Generation**

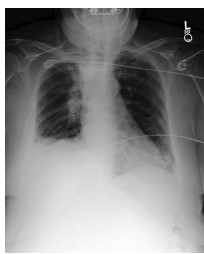

**Ground Truth Report:** (s50043351)
there is a right pleural effusion which is unchanged since prior exam. again seen is a right hilar opacity consistent with fibrosis, better assessed on recent ct. a subtle left lower lobe opacity is seen, which may represent atelectasis, but pneumonia cannot be excluded. the lungs are otherwise clear. the cardiomediastinal silhouette is unchanged from prior exam. visualized osseus structures are unremarkable.

- - - - - - - - - - - - - - - - - - - - - - - - - - - - - - - - - - - - - - -

**Instruction:**
Act as a radiologist and write a diagnostic radiology report for the patient based on their chest radiographs.

- - - - - - - - - - - - - - - - - - - - - - - - - - - - - - - - - - - - - - -

*DeMMo* **Output**
in comparison with study of ___, there is continued enlargement of the cardiac silhouette with evidence for elevated pulmonary venous pressure. opacification along the right lateral chest wall suggests increasing right pleural effusion with compressive atelectasis at the base. mild atelectatic changes are seen at the left base.

---

**Inference Examples of No Context Generation**

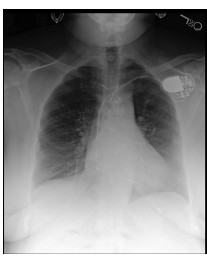

**Ground Truth Report:** (s56104633)
frontal and lateral views of the chest were obtained. patient is status post median sternotomy and cardiac valve replacement. dual lead left-sided pacemaker is seen with leads extending to the expected position of the right atrium and right ventricle. there may be minimal basilar atelectasis. no focal consolidation is seen. there is no pleural effusion or pneumothorax. the cardiac and mediastinal silhouettes are stable and unremarkable.

- - - - - - - - - - - - - - - - - - - - - - - - - - - - - - - - - - - - - - -

**Instruction:**
Based on the given chest X-ray images, please write a diagnostic radiology report for the patient.

- - - - - - - - - - - - - - - - - - - - - - - - - - - - - - - - - - - - - - -

*DeMMo* **Output**
pa and lateral views of the chest provided. left chest wall pacer device is again seen with pacer leads extending to the region of the right atrium and right ventricle. midline sternotomy wires and mediastinal clips are again noted. the lungs are clear without focal consolidation, effusion or pneumothorax. the cardiomediastinal silhouette is normal. imaged osseous structures are intact. no free air below the right hemidiaphragm is seen.

---

**Inference Examples of Generation with Previous Visit as Context**

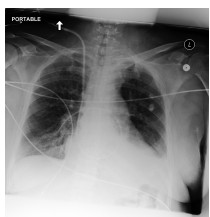

**Ground Truth Report:** (s59751598)
swan-ganz catheter has been advanced beyond the right hilum, and should be withdrawn for standard positioning, as discussed by telephone with dr. ___ at 9:45 a.m. on ___. new airspace opacity distal to the catheter tip could potentially represent pulmonary hemorrhage, but other etiologies such as atelectasis or aspiration are also possible. improving atelectasis in left lower lobe and persistent small left pleural effusion. incidental calcified granulomas within the left upper lobe.

- - - - - - - - - - - - - - - - - - - - - - - - - - - - - - - -

**Context:**
The patient has a previous visit with the report: endotracheal tube and nasogastric tube remain in standard position. swan-ganz catheter has been slightly withdrawn, with tip terminating in the central right hilar region. left sided catheter has been removed, with no visible pneumothorax. cardiac silhouette remains mildly enlarged, but previously reported mild edema has nearly resolved. bibasilar retrocardiac atelectasis is present with some improvement on the left, and small left pleural effusion is unchanged.

- - - - - - - - - - - - - - - - - - - - - - - - - - - - - - - -

**Instruction:**
Considering the patient's previous visit, please write a chest x-ray report for the patient based on the chest x-ray images:

- - - - - - - - - - - - - - - - - - - - - - - - - - - - - - - -

*DeMMo* **Output:**
in comparison with the study of ___, the endotracheal and nasogastric tubes have been removed. swan-ganz catheter remains in place. continued enlargement of the cardiac silhouette with indistinct pulmonary vessels suggesting elevated pulmonary venous pressure. opacification at the left base most likely reflects atelectasis and effusion, though in the appropriate clinical setting, supervening pneumonia would have to be considered. no gross evidence of pneumoperitoneum, though this is not an upright view.

---

**Inference Examples of Generation with Previous Visit as Context**

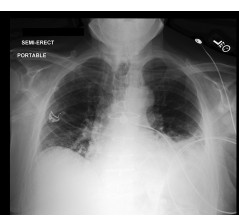

**Ground Truth Report:** (s58402174)
ap portable semi upright view of the chest. lung volumes are low limiting assessment. there is increased bibasilar atelectasis and bronchovascular crowding. overall cardiomediastinal silhouette is unchanged. the right upper extremity access picc line appears in unchanged position extending to the level of the cavoatrial junction. mild congestion is difficult to exclude in the correct clinical setting. no overt signs of edema.

- - - - - - - - - - - - - - - - - - - - - - - - - - - - - - - -

**Context:**
Medical report from the last visit: rounded bilateral mid lung opacities are again seen, grossly unchanged and likely reflect consolidative infectious process given history of septic emboli. there is unchanged bibasilar opacification, which is likely atelectasis with left greater than right effusions. cardiac silhouette is markedly enlarged, similar to the most recent prior. left picc terminates in the cavoatrial junction. median sternotomy wires are intact.

- - - - - - - - - - - - - - - - - - - - - - - - - - - - - - - -

**Instruction:**
Please write a diagnosis chest x-ray report by investigating the given chest x-ray images, referencing the patient's previous report:

- - - - - - - - - - - - - - - - - - - - - - - - - - - - - - - -

*DeMMo* **Output:**

portable semi-upright radiograph of the chest demonstrates low lung volumes which results in bronchovascular crowding. bibasilar atelectasis is unchanged. the cardiomediastinal and hilar contours are unchanged. there is no pneumothorax, pleural effusion, or consolidation. a left-sided picc line ends at the cavoatrial junction. median sternotomy wires are intact.

---

Inference Examples of Generation with Patient's Medical Record as Context

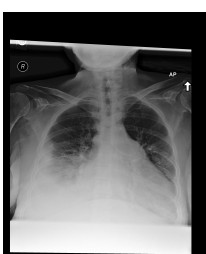

**Ground Truth Report:** (s53025898)

the heart size is moderately enlarged. the mediastinal silhouette and hilar contours are unchanged. a moderate to large right-sided pleural effusion is slightly increased in volume compared to prior examination with collapse of much of the right lower lobe and right middle lobe. there is also some consolidation at the base of the right upper lobe which could be due to compressive atelectasis. there is no left effusion. the upper lung zones appear clear. there is no pneumothorax.

**Context:**

The patient has the following medical conditions and exam result: pulmonary embolism, echocardiogram showing decreased ejection fraction, bronchoscopy revealing inflammation and mucus buildup in the airways.

**Instruction:**

Examine the given chest x-ray images and patient's medical record and write a medical report detailing the findings.

*DeMMo* **Output:**

moderate enlargement of the cardiac silhouette persists. the aorta remains tortuous and diffusely calcified. a moderate to large layering right pleural effusion and possible small left pleural effusion are present, with the right pleural effusion likely larger in size in the interval. lung volumes are low with patchy opacities the lung bases likely reflective of compressive atelectasis. no large pneumothorax is demonstrated. there is no pulmonary vascular congestion.

---

Inference Examples of Generation with Patient's Medical Record as Context

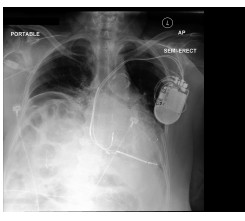

**Ground Truth Report:** (s58585557)

portable semi-upright radiograph of the chest demonstrates low lung volumes with resultant bronchovascular crowding. bibasilar consolidations may represent atelectasis or pneumonia in the appropriate clinical setting. the cardiomediastinal and hilar contours are unchanged. there is a new lucency beneath the right hemidiaphragm concerning for intra-abdominal free air. right-sided picc line and to the mid svc. unchanged position of the aicd. no pneumothorax.

**Context:**

The patient has following information: acute respiratory distress syndrome (ards), chronic obstructive pulmonary disease (copd)

**Instruction:**

Review the attached chest x-ray images and relevant patient information to write a detailed medical report:

*DeMMo* **Output:**
portable semi-upright radiograph of the chest demonstrates low lung volumes with resulting bronchovascular crowding. there is an area of increased opacification at the right lung base, which may represent atelectasis, however aspiration or pneumonia could be considered in the appropriate clinical setting. there is a probable small right-sided pleural effusion. the cardiomediastinal and hilar contours are unchanged. a right-sided internal jugular central venous line ends at the cavoatrial junction. a dual-chamber pacemaker is present, with leads in unchanged position. there is no pneumothorax. free air is seen under the right hemidiaphragm, consistent with peritoneal dialysis.

## Inference Examples of Generation with Template as Context

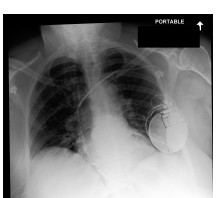

**Ground Truth Report:** (s52736852)
- lung volumes: low
- hemidiaphragm: unchanged chronic elevation of right hemidiaphragm
- focal consolidation: absent
- pulmonary edema: absent
- cardiomediastinal contours: normal
- hilar contours: normal
- pleural effusions: trace, if any, bilateral pleural effusions
- pneumothoraces: absent
- icd: single lead left icd is intact without any lead terminating in the right ventricle.

**Context:**
Template:
- Lung Volumes: [Normal/Low/High/Decreased/Increased]
- Hemidiaphragm: [Normal/Elevated/Depressed/Unchanged]
- Focal Consolidation: [Present/Absent]
- Pulmonary Edema: [Present/Absent]
- Cardiomediastinal Contours: [Normal/Abnormal]
- Hilar Contours: [Normal/Abnormal]
- Pleural Effusions: [Present/Absent/Trace]
- Pneumothoraces: [Present/Absent]
- ICD: [Intact/Damaged/Not Present]

**Instruction:**
Please fill this chest x-ray diagnostic report template based on the give chest x-ray radiographs.

*DeMMo* **Output:**
- lung volumes: low
- hemidiaphragm: unchanged
- focal consolidation: absent
- pulmonary edema: absent
- cardiomediastinal contours: normal
- hilar contours: normal
- pleural effusions: trace bilateral
- pneumothoraces: absent
- icd: intact.

---

Inference Examples of Generation with Template as Context

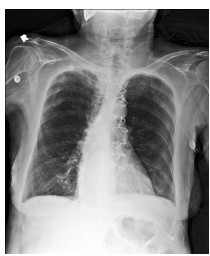

**Ground Truth Report:** (s51791247)
- heart: normal size and no abnormalities noted.
- aorta: diffusely calcified.
- mediastinum: normal contours.
- hilar region: normal.
- lungs: hyperinflated with relative lucency within the lung apices, compatible with emphysema. previously described nodular opacities on ct are not well demonstrated on the current radiograph.
- pleura: no pleural effusion detected.
- pneumothorax: none detected.
- spine: multiple compression deformities of the thoracic spine are unchanged from previous imaging.
- ribs: old bilateral rib deformities.

- - - - - - - - - - - - - - - - - - - - - - - - - - - - - - - - - - - - - - -

**Context:**
Template:
- heart:
- aorta:
- mediastinum:
- hilar region:
- lungs:
- pleura:
- pneumothorax:
- spine:
- ribs:

- - - - - - - - - - - - - - - - - - - - - - - - - - - - - - - - - - - - - - -

**Instruction:**
Write a diagnostic radiology report for the patient based on their chest radiographs following the given template:

- - - - - - - - - - - - - - - - - - - - - - - - - - - - - - - - - - - - - - -

*DeMMo* **Output:**
- heart: normal in size.
- aorta: calcified.
- mediastinum: unchanged.
- hilar region: unchanged.
- lungs: hyperinflated.
- pleura: no pleural effusion or pneumothorax.
- pneumothorax: absent.
- spine: incompletely characterized cervical spinal fusion appears unchanged.
- ribs: healed right posterolateral sixth and seventh rib fractures appear unchanged.

---

Inference Examples of Report Revision

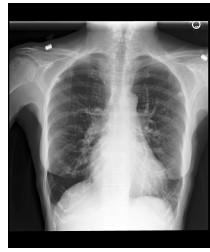

**Ground Truth Report:** (s57977763)
frontal and lateral views of the chest were obtained. the lungs are hyperinflated/well expanded. costochondral calcification is noted. no definite focal consolidation is seen. there is no pleural effusion or pneumothorax. the cardiac and mediastinal silhouettes are stable.

**Context:**
Incorrect report: Frontal and lateral views of the chest were obtained. The lungs are hyperinflated/well expanded. Costochondral calcification is noted. There is a small area of focal consolidation in the left upper lobe. No pleural effusion or pneumothorax is seen. The cardiac and mediastinal silhouettes are stable.

---

**Instruction:**
Remove the diagnosis of acute focal consolidation

---

*DeMMo* **Output:**
frontal and lateral views of the chest were obtained. the lungs are hyperinflated well expanded. costochondral calcification is noted. there is no focal consolidation, pleural effusion or pneumothorax. the cardiac and mediastinal silhouettes are stable.

---

### Inference Examples of Report Revision

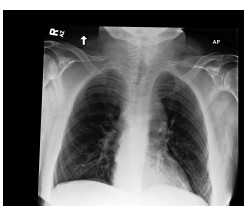

**Ground Truth Report:** (s56541072)
heart size is normal. the aorta is tortuous. unchanged widening of the mediastinum attributable to mediastinal lipomatosis is re- demonstrated. hilar contours are unremarkable. pulmonary vasculature is not engorged. lungs are clear. no pleural effusion, focal consolidation or pneumothorax is demonstrated. there are no acute osseous abnormalities.

---

**Context:**
Report: heart size is enlarged. the aorta is tortuous. unchanged widening of the mediastinum attributable to mediastinal mass is re-demonstrated. hilar contours are unremarkable. pulmonary vasculature is engorged. lungs show bilateral infiltrates. a small pleural effusion is noted on the right. no acute osseous abnormalities.

---

**Instruction:**
Update the medical report of the given chest x-ray images with these changes: heart size is normal. the mediastinum is widened but it is not due to mediastinal mass, instead, it is due to mediastinal lipomatosis. the pulmonary vasculature is not engorged. the lungs are clear without any infiltrates or pleural effusion.

---

*DeMMo* **Output:**
heart size is normal. the aorta is tortuous. unchanged widening of the mediastinum attributable to mediastinal lipomatosis is re- demonstrated. hilar contours are unremarkable. pulmonary vasculature is not engorged. lungs are clear without focal consolidation. no pleural effusion or pneumothorax is present. there are no acute osseous abnormalities.

## E  COMPARISON WITH OTHER METHODS

---

**Inference Examples of Generation with No Context**

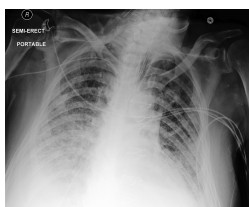

**Ground Truth Report:** (s55575670)

the patient has a history of chronic interstitial lung disease with waxing and waning pulmonary edema and infection. today it is largely unchanged with diffuse infiltrative and interstitial opacities stable since ___. bilateral pleural effusion is essentially the same. cardiomediastinal silhouette is stable and demonstrates mild cardiomegaly. there is no pneumothorax. enteric tube is seen once again, entering the stomach and then out of field of view. right-sided picc terminates within the mid svc. an endotracheal tube terminates no less than 6 cm from the carina.

---

**Instruction:**
Act as a radiologist and please write a chest x-ray report for the patient based on the chest x-ray images:

---

**ChatCAD+**
upon reviewing the patient's chest x-ray, it has been observed that overlying ekg leads are present, and lung volumes are low, limiting assessment. there is diffuse pulmonary edema with probable small bilateral pleural effusions, and it is difficult to exclude a superimposed pneumonia. the heart size cannot be assessed due to the presence of these conditions, and the mediastinal contour appears grossly unremarkable. the bony structures are intact. while there is no sign of cardiomegaly, further evaluation is necessary to investigate potential consolidation and atelectasis, and to monitor the patient's pulmonary edema and pleural effusion.

---

**CvT212DistilGPT2**
as compared to the previous radiograph there is no relevant change. the monitoring and support devices are constant. the bilateral parenchymal opacities are constant in extent and severity. unchanged appearance of the cardiac silhouette.

---

*DeMMo* **(Ours):**
in comparison with the study of ___, the monitoring and support devices remain in place. continued enlargement of the cardiac silhouette with pulmonary edema. bibasilar opacifications persist, consistent with a combination of pleural effusion and atelectasis. The patient has no pneumothorax. in the appropriate clinical setting, supervening pneumonia would have to be considered.

---

**Inference Examples of Generation with No Context**

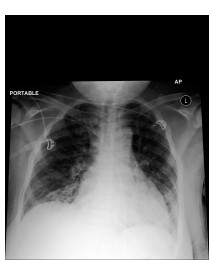

**Ground Truth Report:** (s50324889)
- heart size: mild to moderately enlarged
- mediastinal contour: unchanged
- pulmonary edema: mild to moderate, worse in the interval
- lung bases: patchy opacities likely reflect areas of atelectasis
- atelectasis: present in the lung bases
- pleural effusion: trace right pleural effusion
- pneumothorax: none identified
- osseous abnormalities: none detected.

---

**Context:**
Template:
- heart size:

- mediastinal contour:
- pulmonary edema:
- lung bases:
- atelectasis:
- pleural effusion:
- pneumothorax:
- osseous abnormalities:

---

**Instruction:**
Write a diagnostic radiology report for the patient based on their chest radiographs following the given template:

---

**ChatCAD+**
i'm sorry, but as an ai language model, i do not have access to the patient's chest x-ray image and cannot provide a report based on it. however, based on the ai reports, the findings suggest that the patient has a mildly enlarged heart mentioned in network b, and there is a small possibility of cardiomegaly, consolidation, or pleural effusion mentioned in network a. the patient is likely to have edema or atelectasis mentioned in network a. mediastinal contours are unremarkable mentioned in network b. crowding of bronchovascular structures is present without overt pulmonary edema mentioned in network b. patchy opacities in the lung bases likely reflect areas of atelectasis mentioned in network b. no large pleural effusion or pneumothorax is present mentioned in network b. in the report, the presence or absence of heart size, mediastinal contour, pulmonary edema, lung bases, atelectasis, pleural effusion, pneumothorax, and osseous abnormalities should be noted, if present. however, it is important to note that ai reports are not a substitute for a thorough clinical evaluation by a medical professional. the patient's medical history and clinical presentation should be taken into consideration to determine the significance of the findings on the chest x-ray. additional imaging studies or laboratory tests may be necessary to determine the underlying etiology, if any.

---

**CvT212DistilGPT2**
- lung fields
- pulmonary edema
- severity moderate
- distribution diffuse
- consolidation
- location none
- distribution an
- pleural spaces
- effusion
- location none
- size na
- pneumothorax
- location none
- size na
- cardiac silhouette
- size mild cardiomegaly
- shape na
- mediastinum
- contour stable
- lymph nodes na

---

*DeMMo* **(Ours):**
- heart size: moderately enlarged
- mediastinal contour: normal
- pulmonary edema: mild
- lung bases: patchy opacities likely reflecting atelectasis
- atelectasis: present in the lung bases
- pleural effusion: right side
- pneumothorax: none identified
- osseous abnormalities: none identified.

Comparing ours with the baseline approaches, ChatCAD+ usually gives relatively accurate report, but the output is usually long and contains a lot of non-informative sentences. Non-LLM encoder-decoder model underperforms in terms of clinical efficacy. Both of these two baselines usually fail to understand the input context by either ignoring it or following a wrong template. In contrast, our method produces accurate and concise output, and can follow the instruction and context accurately.

## F    MORE ON DATA VALIDATION

**Quality analysis**    To further assess the quality of generated dataset, this time we sampled 100 more cases of all tasks for a manual validation by a medical professional. We update this result in the appendix of the paper. The data quality for samples is shown in table 7.

| Type | Number |
|---|---|
| Acceptable | 74 |
| Tolerable | 11 |
| Unacceptable | 15 |
| Total Samples | 100 |

Table 7: Result of data quality manual evaluation.

- Acceptable: Out of the samples inspected, 74 were deemed to be of good quality. Both instructions and generated contents were accurate.
- Tolerable: Eleven cases were considered tolerable and carried no factual mistakes. For instance:
    - A case with a positive pollen allergy test in the prediction of medical history and exam results without any indications from the original report.
    - A case generated a template that did not have accurate indentation.
- Unacceptable: Fifteen cases were identified as potentially containing mistakes and were considered unacceptable. Some cases with incorrect generated instructions or contents. A few cases that generated reports not following the provided template.

**Fully validated test set**    Besides validating the generated data, we have validated a small-scale test dataset including all the real-world tasks mentioned in our paper with 20 samples for each task, total of 100 samples. Based on the ground truth medical report, we let a specialist validate or manually create report template data and report correction data. We use the MIMIC-CXR ground truth data with previous report, and ground truth reports that have an indication section (reason of examination) to serve as ground truth medical history/condition context data. The results obtained from testing on this validated test set serve as supplementary findings to the test results. On this fully validated test set, the performance of our trained model and ChatCAD+ baseline is shown in table 8. The result shows no significant deviation from our reported result on generated test set and outperforms the ChatCAD+ baseline on F1 score as well, which shows that our model trained on generated data performs well on real validated test set as well, which indirectly proves that the gap between generated data and real validated data is small.

**Statistical validation and significance testing**    We select the ChatCAD+ model as a baseline to conduct statistical significance test against ours, across five tasks on multiple metrics. The Precision, Recall and F1 metrics are not included in the comparison because they are composite indicators for multi-classification and are not applicable in this case. BLEU@1, BLEU@2, BLEU@3, BLEU@4, METEOR and ROUGE-L metrics are computed 95% confidence interval and tested. The test sample size is substantial, and the differences are quite noticeable. All these metric differences showed statistically significant, with p-values less than 0.001. The detailed comparisons are shown in table 9.

**Large scale validation**    Medical domain data is more sensitive and therefore needs careful validation by medical specialists. As we mentioned in section 4.3, we invite medical professionals to validate our generated data and find that most generated data are consistent with the ground truth

| Task | Method | B@1 | B@2 | B@3 | B@4 | M | R-L | P | R | F1 |
|------|--------|-----|-----|-----|-----|---|-----|---|---|-----|
| No Context | ChatCAD+ | 0.312 | 0.152 | 0.074 | 0.043 | 0.283 | 0.186 | 0.385 | **0.500** | 0.418 |
| | Ours | **0.404** | **0.257** | **0.178** | **0.133** | **0.327** | **0.278** | **0.475** | 0.380 | **0.422** |
| Revision | ChatCAD+ | 0.828 | 0.779 | 0.730 | 0.690 | 0.838 | 0.803 | 0.721 | 0.936 | 0.814 |
| | Ours | **0.929** | **0.913** | **0.898** | **0.883** | **0.931** | **0.939** | **0.978** | **0.957** | **0.967** |
| Template | ChatCAD+ | 0.274 | 0.218 | 0.178 | 0.143 | 0.489 | 0.270 | 0.471 | **0.825** | 0.600 |
| | Ours | **0.634** | **0.543** | **0.471** | **0.412** | **0.637** | **0.565** | **0.892** | 0.805 | **0.846** |
| Previous Report | ChatCAD+ | 0.302 | 0.174 | 0.113 | 0.079 | **0.334** | 0.215 | 0.438 | **0.471** | 0.454 |
| | Ours | **0.343** | **0.213** | **0.143** | **0.107** | 0.287 | **0.250** | **0.500** | 0.449 | **0.478** |
| Medical Record | ChatCAD+ | 0.166 | 0.094 | 0.055 | 0.035 | 0.028 | 0.151 | 0.282 | **0.511** | 0.364 |
| | Ours | **0.377** | **0.218** | **0.134** | **0.087** | **0.299** | **0.245** | **0.459** | 0.395 | **0.425** |

Table 8: Comparison of our model with ChatCAD+ on the fully validated test sets.

| Task | Method | B@1 | B@2 | B@3 | B@4 | M | R-L |
|------|--------|-----|-----|-----|-----|---|-----|
| No Context | ChatCAD+ | 0.292 (0.289,0.296) | 0.154 (0.151,0.157) | 0.082 (0.079,0.084) | 0.041 (0.038,0.043) | 0.278 (0.275,0.282) | 0.194 (0.191,0.196) |
| | Ours | 0.326 (0.321,0.332) $*p < 0.001$ | 0.197 (0.192,0.202) $*p < 0.001$ | 0.115 (0.110,0.120) $*p < 0.001$ | 0.070 (0.065,0.074) $*p < 0.001$ | 0.290 (0.285,0.295) $*p < 0.001$ | 0.239 (0.234,0.243) $*p < 0.001$ |
| Revision | ChatCAD+ | 0.521 (0.512,0.530) | 0.459 (0.450,0.469) | 0.414 (0.405,0.424) | 0.378 (0.368,0.388) | 0.700 (0.692,0.708) | 0.556 (0.547,0.566) |
| | Ours | 0.891 (0.884,0.898) $*p < 0.001$ | 0.868 (0.860,0.875) $*p < 0.001$ | 0.847 (0.839,0.855) $*p < 0.001$ | 0.828 (0.820,0.837) $*p < 0.001$ | 0.905 (0.898,0.911) $*p < 0.001$ | 0.902 (0.896,0.909) $*p < 0.001$ |
| Template | ChatCAD+ | 0.304 (0.296,0.312) | 0.246 (0.238,0.253) | 0.204 (0.197,0.211) | 0.169 (0.162,0.176) | 0.450 (0.441,0.458) | 0.285 (0.277,0.292) |
| | Ours | 0.621 (0.610,0.632) $*p < 0.001$ | 0.555 (0.543,0.567) $*p < 0.001$ | 0.502 (0.489,0.514) $*p < 0.001$ | 0.455 (0.442,0.468) $*p < 0.001$ | 0.631 (0.620,0.642) $*p < 0.001$ | 0.605 (0.594,0.616) $*p < 0.001$ |
| Previous Report | ChatCAD+ | 0.241 (0.236,0.245) | 0.134 (0.130,0.137) | 0.076 (0.073,0.079) | 0.042 (0.039,0.045) | 0.304 (0.299,0.308) | 0.179 (0.176,0.182) |
| | Ours | 0.298 (0.290,0.306) $*p < 0.001$ | 0.181 (0.174,0.187) $*p < 0.001$ | 0.106 (0.099,0.112) $*p < 0.001$ | 0.064 (0.058,0.070) $*p < 0.001$ | 0.276 (0.269,0.283) $*p < 0.001$ | 0.236 (0.230,0.242) $*p < 0.001$ |
| Medical Record | ChatCAD+ | 0.139 (0.135,0.143) | 0.061 (0.059,0.064) | 0.022 (0.020,0.024) | 0.010 (0.008,0.011) | 0.206 (0.201,0.210) | 0.108 (0.106,0.111) |
| | Ours | 0.302 (0.291,0.313) $*p < 0.001$ | 0.183 (0.174,0.191) $*p < 0.001$ | 0.108 (0.101,0.115) $*p < 0.001$ | 0.065 (0.058,0.071) $*p < 0.001$ | 0.275 (0.266,0.284) $*p < 0.001$ | 0.235 (0.228,0.242) $*p < 0.001$ |

Table 9: Statistical significance test of our model compared to ChatCAD+ on the NLG metrics.

radiology report without factual errors. Due to the time constraint, we only have a small random sampled subset that undergoes clinical validation. We will have more human validations, including large-scale validation through crowdsourcing medical professionals, in the future before we release the data.

# G   LIMITATIONS AND FUTURE WORKS

As introduced in previous sections, our method is a pure generation method without encompassing extra generation priors such as labels from a classifier. In contrast, methods such as Zhao et al. (2023) and You et al. (2021) utilize an image classifier to extract disease labels prior to generation, which ensures diagnostic correctness. Tanida et al. (2023) leverages object detector and use the extracted abnormal regions to guide generation, which also shows promising result. This presents a limitation, as our model's diagnostic accuracy may not be as reliable as methods employing guidance from high accuracy classifiers. Therefore, future works may focus on fusing the model with extra generation prior or guidance to further improve clinical efficacy.

We also observe that our *DeMMo* approach can be generalized to other domains as well using other domain-specific vision encoders. A potential future direction could entail utilizing a CT scan encoder for CT report generation, or developing a universal medical vision encoder for a more unified medical report generation tasks.

