# OpenReview forum: "Large Multimodal Model for Real-World Radiology Report Generation"
_ICLR.cc/2024/Conference — Submitted to ICLR 2024_

### Official Review · Reviewer_uSN9 · 2023-10-31

**Soundness:** 3 good
**Presentation:** 4 excellent
**Contribution:** 3 good
**Rating:** 8
**Confidence:** 2

**Summary:**

This paper introduced a new problem setting, real-world radiology report generation, that focuses on interactivity, following instructions, and considering various context information. The authors constructed a new benchmark dataset for this purpose and proposed a Domain-enhanced Multi-Modal (DeMMo) model, a variant of Flamingo model, to improve the medical domain-specific abilities.

**Strengths:**

- The focus on real-world radiology report generation is designed to be practical and applicable in clinical settings, which is great in terms of translational impact.
- The benchmark is very useful to the community.
- The model integrates an additional domain-specific medical encoder to the perceiver resampler, enhancing its ability to capture detailed visual features in the medical domain.

**Weaknesses:**

- Lack of model efficiency analysis
- Since the proposed benchmark is one of the core contributions in this work, please describe how do you plan to make it publicly available.

**Questions:**

1. Could you please add the number of model parameters and flops in the result Tables?
2. How do you plan to make the dataset available to the community?
3. Where do you plan to host this benchmark? CodaLab could be a good platform.

---

> ### Author Response · Authors · 2023-11-22
> **Response to Reviewer uSN9**
>
> We thank you for your time and valuable comments.
>
> ### Response to model efficiency (Weakness 1 and Question 1)
>
> > Lack of model efficiency analysis
> > Could you please add the number of model parameters and flops in the result Tables?
>
> Since the baseline ChatCAD+ employs ChatGPT API for summarization, it makes the comparison of model parameters or FLOPs not applicable. For our model, the total number of parameters is about 8.25 billion, and the FLOPs is about 1,000 to 2000 billion, depending on the length of input and out.
>
> ### Response to data publication (Weakness 2 & Question 2 & 3)
>
> > Since the proposed benchmark is one of the core contributions in this work, please describe how do you plan to make it publicly available.
> > How do you plan to make the dataset available to the community?
> > Where do you plan to host this benchmark? CodaLab could be a good platform.
>
> We plan to release our dataset on PhysioNet (where MIMIC-CXR is hosted) after a larger-scale validation and cleaning.

---

### Official Review · Reviewer_watP · 2023-11-01

**Soundness:** 1 poor
**Presentation:** 1 poor
**Contribution:** 1 poor
**Rating:** 3
**Confidence:** 5

**Summary:**

The authors introduce a novel problem setting for real-world report generation that closely simulates clinical practices by integrating various clinical interactions and contextual information. They also present DeMMo, a substantial multimodal model enriched with domain-specific capabilities achieved by integrating a general domain Flamingo model with an additional medical vision encoder.

**Strengths:**

(1) The incorporation of user instructions into the generation process is a valuable enhancement for improving the quality of generated reports.

**Weaknesses:**

(1) The novelty of DeMMo is somewhat limited.
(2) The inclusion of user instructions may raise concerns about the trustworthiness of generated texts, necessitating careful manual review by doctors, potentially leading to increased time and effort.
(3) The test dataset sizes are notably smaller in comparison to the training datasets.
(4) Relying solely on a single self-constructed dataset for experiments lacks robustness. Additional datasets should be considered for validation.
(5) The addition of an extra encoder in DeMMo could potentially introduce computational overhead, impacting overall efficiency.

**Questions:**

(1) In Table 5, DeMMo's performance on BLEU@1 appears suboptimal. Could you explain the reasons for this lower performance?

---

> ### Author Response · Authors · 2023-11-22
> **Response to Reviewer watP**
>
> We appreciate your valuable reviews and comments.
> ### Response to novelty (Weakness 1)
>
> > The novelty of DeMMo is somewhat limited.
>
> Please refer to the discussion on novelty in our general response.
>
> ### Response to data validation (Weakness 2)
>
> > The inclusion of user instructions may raise concerns about the trustworthiness of generated texts, necessitating careful manual review by doctors, potentially leading to increased time and effort.
>
> Please refer to our general response for discussion on data validation.
>
> ### Response to test data size (Weakness 3)
>
> > The test dataset sizes are notably smaller in comparison to the training datasets.
>
> Our dataset is generated based on MIMIC-CXR, therefore we adopt the official split of the dataset, which means all data generated using test samples in MIMIC-CXR is also used as test sample in our generated data. After some automatic parsing and filtering, the generated test set size reduces accordingly with generated training set. Therefore we think our test set size is reasonable by referring to the official train/test ratio of MIMIC-CXR.
>
> ### Response to additional datasets (Weakness 4)
>
> > Relying solely on a single self-constructed dataset for experiments lacks robustness. Additional datasets should be considered for validation.
>
> - **MIMIC-CXR** We only conduct experiment on MIMIC-CXR because it is the most popular and largest high-quality report generation dataset to date. Other report generation datasets such as IU X-Ray has only about 3,000 samples. We think setting up another benchmark and generating a set of context data on such small-scale dataset would not give much benefits or significance.
> - **Additional validation** Since we propose a new set of real-world report generation tasks with additional context, we could only rely on the self-constructed dataset for experiment, which also includes the real MIMIC-CXR dataset. For additional validation dataset, we have constructed a fully validated test set for further validation of our model trained on the generated dataset. Please refer to the response to data validation section in our general response.
>
>
> ### Response to computational overhead (Weakness 5)
>
> > The addition of an extra encoder in DeMMo could potentially introduce computational overhead, impacting overall efficiency.
>
> Compared to the LLM part of Flamingo, the number of paramters in the newly added vision encoder, which is a ResNet-50 model, is very small. The Flamingo model has more than 8 billions of parameters, while the newly added vision encoder only has about 26 million parameters, which is about 0.3% more parameters. The complexity increase can be ignored.
>
> ### Response to lower BLEU@1 (Question 1)
>
> > In Table 5, DeMMo's performance on BLEU@1 appears suboptimal. Could you explain the reasons for this lower performance?
>
> - **CE metrics preferred over NLG metrics.** We want to first emphasize the significance of clinical efficacy (precision, recall, F1) over NLG metrics for report generation tasks. N-gram-based NLG metrics such as BLEU generally enforce a strict word matching and therefore lack semantic understanding. In comparison, CE metrics employ robust NLP tools to extract pathological labels from text for comparison, which is more crucial for clinal report tasks. It is possible for a generated report that has lots of word overlap with ground truth but no correct clinical findings.
> - **Reason for lower BLEU@1** This means our model generates text that, while perhaps not closely matching the ground truth report in terms of phrasing or word choice, is however more effective in a real-world clinical context. This shows that our model is able to generalize better and give more diverse phrases while also maintaining good clinical efficacy, but the baseline methods may be overfitted to some high-frequency phrases or sentences that do not give enough useful information (e.g. frontal and lateral views of the chest were obtained) and therefore does not give accurate clinical diagnosis.

---

### Official Review · Reviewer_8G3W · 2023-11-03

**Soundness:** 2 fair
**Presentation:** 3 good
**Contribution:** 2 fair
**Rating:** 5
**Confidence:** 4

**Summary:**

This paper propose a new benchmark dataset for medical report generation with building a unified data generation pipeline. It also proposed a domain-specific multi-modal model (DeMMo) to improve the raw llm for medical report generation. Experiments show the method is good.

**Strengths:**

1. The motivation makes sense. It should be quite helpful and natural to use instruction/context to generate better medical report.
2. The paper has did plenty of experiments to show the effectiveness of the proposed method.

**Weaknesses:**

1. It is not clear how to generate the I, C and R', which is critical in this paper.  Also, I'm not sure if the quality of generated data by the unified pipeline is good or not, though the authors mention there are professions who help check them.
2. The comparison in Table 3 shows the advantage of the proposed method in this paper, which is mainly due to the domain-specific encoder. However, will the computational complexity be much larger?
3. The dataset (MIMIC-R3G) is not open sourced or not mentioned to open source it in future.

**Questions:**

1. I'm not sure if the generated I, C and R' are fixed or can be different at different time. To me, a benchmark dataset should better be fixed.
2. Does every patient have the previous visit data in MIMIC-CXR? How do you deal with those who have no previous data?
3. "In summary, our medical professionals have determined that no significant factual discrepancies exist between the content generated by GPT and the ground-truth reports across all samples". Can you please introduce more about it? Especially considering the huge number of samples in the dataset.
4. The experiments mainly talk about the effectiveness of the method, how we can assess the quality of the data.

**Details Of Ethics Concerns:**

Considering the R3G is generated by a pipeline, so we are not sure about if there are discrimination/bias/fairness concerns.

---

> ### Author Response · Authors · 2023-11-22
> **Response to Reviewer 8G3W**
>
> We appreciate your valuable comments.
>
> ### Response to generated context (Weakness 1 & Question 1)
>
> > It is not clear how to generate the I, C and R', which is critical in this paper...
> > I'm not sure if the generated I, C and R' are fixed or can be different at different time. To me, a benchmark dataset should better be fixed.
>
>  - **Generating context** Due to limited space, we show more examples of how to generate context data using a ground truth report in Appendix B. In general, we design specific system messages for each task and input the ground truth report to ChatGPT, with some in-context examples to guide the format of response. We then parse the formatted ChatGPT response as the context data $I$, $C$, and $R'$ for each task respectively.
>  - **The benchmark is fixed** It is true that ChatGPT introduces randomness in the generation result, however, the benchmark dataset should be the generated result, which is fixed. Many other outstanding works follow the same paradigms for creating datasets/benchmarks using GPT or similar LLMs [1,2,3,4,5].
>
> ### Response to computational complexity (Weakness 2)
>
> > The comparison in Table 3 shows the advantage of the proposed method in this paper, which is mainly due to the domain-specific encoder. However, will the computational complexity be much larger?
>
> Compared to the LLM part of Flamingo, the number of parameters in the newly added vision encoder, which is a ResNet-50 model, is very small. The Flamingo model has more than 8 billion parameters, while the newly added vision encoder only has about 26 million parameters, which is about 0.3% increase. The complexity increase can be ignored.
>
> ### Response to data publication (Weakness 3)
>
> > The dataset (MIMIC-R3G) is not open sourced or not mentioned to open source it in future.
>
> We plan to release our dataset on PhysioNet (where MIMIC-CXR is hosted) after a larger-scale validation and cleaning.
>
> ### Response to previous visit data (Question 2)
>
> > Does every patient have the previous visit data in MIMIC-CXR? How do you deal with those who have no previous data?
>
> No, not all patients have a previous visit data in MIMIC-CXR. In our experiments on MIMIC-CXR, we simply exclude the sample for this task in the dataset when the previous data is not available.
>
> ### Response to data validation (Weakness 1 & Question 3 & 4)
>
> > ...Also, I'm not sure if the quality of generated data by the unified pipeline is good or not, though the authors mention there are professions who help check them.
> > "In summary, our medical professionals have determined that no significant factual discrepancies exist between the content generated by GPT and the ground-truth reports across all samples". Can you please introduce more about it? Especially considering the huge number of samples in the dataset.
> > The experiments mainly talk about the effectiveness of the method, how we can assess the quality of the data.
>
> Please refer to our general response on dataset validation.
>
> ### Response to ethics concerns
>
> > Considering the R3G is generated by a pipeline, so we are not sure about if there are discrimination/bias/fairness concerns.
>
> Our generated data is based on MIMIC-CXR, which undergoes appropriate anonymization process, with all sensitive information that could potentially lead to discrimination/bias/fairness issues being masked. We believe our generation approach, specifically on generating medical record/test results, is neutral and will not introduce extra ethical concerns given that the input has no source of discrimination/bias such as race, nationality, social status, etc., and only focus on medical reasoning. Other tasks will not introduce discrimination/fairness issues since we only do generation at format or text level modification.
>
> Reference:
>
> [1] Liu, Haotian, et al. "Visual instruction tuning." arXiv preprint arXiv:2304.08485 (2023).
>
> [2] Li, Bohao, et al. "Seed-bench: Benchmarking multimodal llms with generative comprehension." arXiv preprint arXiv:2307.16125 (2023).
>
> [3] Yin, Zhenfei, et al. "LAMM: Language-Assisted Multi-Modal Instruction-Tuning Dataset, Framework, and Benchmark." arXiv preprint arXiv:2306.06687 (2023).
>
> [4] Li, Bo, et al. "Mimic-it: Multi-modal in-context instruction tuning." arXiv preprint arXiv:2306.05425 (2023).
>
> [5] Li, KunChang, et al. "Videochat: Chat-centric video understanding." arXiv preprint arXiv:2305.06355 (2023).

---

### Official Review · Reviewer_GGyY · 2023-11-05

**Soundness:** 3 good
**Presentation:** 3 good
**Contribution:** 2 fair
**Rating:** 5
**Confidence:** 4

**Summary:**

The paper proposes an automated radiology report generation system trained on the newly created MIMIC-R3G dataset. This dataset focuses on practical tasks that are somewhat overlooked in the literature. The authors employ a GPT-based data generation pipeline to synthesize training data across different R3G tasks. The experimental outcomes presented in the paper suggest that the proposed DeMMo can outperform existing approaches in radiology report generation according to evaluations on the MIMIC-R3G benchmark.

**Strengths:**

- The manuscript targets meaningful and less explored tasks in radiology report generation, covering a variety of scenarios. This focus is commendable as it moves the field towards clinically applicable solutions in automatic report generation.
- By utilizing ChatGPT for data generation, the authors propose a potential solution to mitigate the issue of data scarcity in complex report generation tasks.
- The adaptation of Flamingo and prompt tuning in the proposed DeMMo model demonstrate improved quantitative results over existing methods.
- The manuscript is well-written and easy to follow.

**Weaknesses:**

- Despite addressing significant tasks in clinical practice, the manuscript's technical contributions appear incremental and may not align with the expectations of the ICLR community. The content might be more suited to specialized medical-related conferences like MICCAI or IPMI.
- The methodology for constructing the MIMIC-R3G dataset, particularly the generation of reports that integrate prior patient visits and additional lab test information, lacks rigorous clinical validation. To ensure the clinical relevance of the generated data, it would be beneficial to include samples that have been validated by healthcare professionals. This is also true for the generated reports.
- Detailed statistical validation and significance testing could strengthen the results section. While the proposed DeMMo generally performs better than baseline models in terms of NLG metrics, the recall of CE seems to be lower than ChatCAD.
- The paper would benefit from a more in-depth qualitative analysis. Comparative examples of reports generated by different models could offer valuable insights into each model's strengths and limitations, providing a clearer understanding of the practical implications of their use in clinical settings.

**Questions:**

- The manuscript would greatly benefit from the inclusion of clinical validations for the synthesized training data. Can the authors present any evaluations conducted by medical professionals to verify the clinical accuracy of the generated data? The same question applies to the generated reports; providing clinical validations would significantly enhance the paper's credibility.
- Will the constructed MIMIC-R3G benchmark be publically available?
- I would recommend a more thorough statistical analysis of the results to better elucidate the significance of the findings. Additionally, providing qualitative comparisons of the reports generated by DeMMo and other baseline models would offer deeper insights into the practical utility of the proposed model.
- It is noted that the recall for CE by DeMMo tends to be lower compared to the baseline model ChatCAD. Could the authors delve into possible reasons for this discrepancy and suggest potential improvements?

Minor:
- Citation formats in 6.2 and Supp: A are inconsistent. There is one missing citation in Supp: E.

---

> ### Author Response · Authors · 2023-11-22
> **Response to Reviewer GGyY (Part 1)**
>
> Thank you for the valuable review and comments.
>
> ### Response to technical contribution (Weakness 1)
>
> > Despite addressing significant tasks in clinical practice, the manuscript's technical contributions appear incremental and may not align with the expectations of the ICLR community. The content might be more suited to specialized medical-related conferences like MICCAI or IPMI.
>
> Please refer to our general response. Also, there are many other medical domain works that have been accepted or received good feedback from the reviewers on ICLR last year [1,2,3] and this year [4,5]. Moreover, our proposed framework DeMMo can be extended to enhance multimoal LLM for other domains as well. Therefore we think our paper should also be valuable for the ICLR community.
>
> ### Response to data validation (Weakness 2 & Question 1)
>
> > The methodology for constructing the MIMIC-R3G dataset, particularly the generation of reports that integrate prior patient visits and additional lab test information, lacks rigorous clinical validation. To ensure the clinical relevance of the generated data, it would be beneficial to include samples that have been validated by healthcare professionals. This is also true for the generated reports.
> > The manuscript would greatly benefit from the inclusion of clinical validations for the synthesized training data. Can the authors present any evaluations conducted by medical professionals to verify the clinical accuracy of the generated data? The same question applies to the generated reports; providing clinical validations would significantly enhance the paper's credibility.
>
> Please refer to our general response. In our experiments we do not use generated report for prior visits since we can retrieve prior visits within the MIMIC-CXR data following metadata included. For 4 of the 5 tasks, we directly use the ground truth from MIMIC-CXR dataset unchanged. The template task uses the generated ground truth, but the modification should be only on the format scale and does not affect the clinical information. We have validated on small subset and we will perform further larger-scale validation and cleaning before releasing the dataset.
>
> ### Response to statistical analysis (Weakness 3 & Question 3)
>
> > Detailed statistical validation and significance testing could strengthen the results section. While the proposed DeMMo generally performs better than baseline models in terms of NLG metrics...
> > I would recommend a more thorough statistical analysis of the results to better elucidate the significance of the findings...
>
> Please refer to Part 3 of our general response.
>
> ### Response to more qualitative results (Weakness 4 & Question 3)
>
> > The paper would benefit from a more in-depth qualitative analysis. Comparative examples of reports generated by different models could offer valuable insights into each model's strengths and limitations, providing a clearer understanding of the practical implications of their use in clinical settings.
> > ...Additionally, providing qualitative comparisons of the reports generated by DeMMo and other baseline models would offer deeper insights into the practical utility of the proposed model.
>
> We add an additional section in the appendix to show some comparison results of ours and baseline output and provide some analysis on the strengths and limitations of each method.
>
> ### Response to lower recall (Weakness 3 & Question 4)
>
> > While the proposed DeMMo generally performs better than baseline models in terms of NLG metrics, the recall of CE seems to be lower than ChatCAD.
> > It is noted that the recall for CE by DeMMo tends to be lower compared to the baseline model ChatCAD. Could the authors delve into possible reasons for this discrepancy and suggest potential improvements?
>
> ChatCAD+ is a framework that leverages a pretrained classifier to produce some initial diagnosis result based on the input image. The pretrained classifier is on 5 pathologic labels only but the accuracy is guaranteed and employed extra classification training data, while our approach does not have such accurate prior information or extra training data therefore a slightly lower recall in some cases. For future works we might work on incorporating additional models for combining results from different modalities as well including classification results and grounding/segmentation results.
>
>
> ### Response to dataset publication (Question 2)
>
> > Will the constructed MIMIC-R3G benchmark be publically available?
>
> Yes, we plan to release our dataset on PhysioNet (where MIMIC-CXR is hosted) after a larger-scale validation and cleaning.

---

> ### Author Response · Authors · 2023-11-22
> **Response to Reviewer GGyY (Part 2)**
>
> ### Response to Minor
>
> > Citation formats in 6.2 and Supp: A are inconsistent. There is one missing citation in Supp: E.
>
> Thank you so much for pointing these out. We have fixed the citations.
>
>
> Reference:
>
> [1] Qin, Ziyuan, et al. "Medical image understanding with pretrained vision language models: A comprehensive study." The Eleventh International Conference on Learning Representations (2023).
>
> [2] Zong, Yongshuo, et al. "MEDFAIR: Benchmarking Fairness for Medical Imaging." The Eleventh International Conference on Learning Representations (2023).
>
> [3] Zhao, Yuzhong, et al. "AE-FLOW: Autoencoders with Normalizing Flows for Medical Images Anomaly Detection." The Eleventh International Conference on Learning Representations (2023).
>
> [4] Anonymous. "MedJourney: Counterfactual Medical Image Generation by Instruction-Learning from Multimodal Patient Journeys." Submitted to The Twelfth International Conference on Learning Representations (2023).
>
> [5] Anonymous. "LLM-CXR: Instruction-Finetuned LLM for CXR Image Understanding and Generation." Submitted to The Twelfth International Conference on Learning Representations (2023).

---

### Official Review · Reviewer_NXpC · 2023-11-20

**Soundness:** 2 fair
**Presentation:** 3 good
**Contribution:** 2 fair
**Rating:** 5
**Confidence:** 4

**Summary:**

This paper made serveral technical contributions to mimic how a real radiologist will do the report generation process in a more realistic work environment. All these contributions as listed in section one are valid scientific contributions. I think there should be some credits for that, advancing from the previous literature.

This paper/work is developed using Flemingo general vision encoder and ChatGPT tools. This would be reasonable but the dependency on using ChatGPT will decrease its scientific values. More detailed comments will follow in later comments.

**Strengths:**

This paper made serveral technical contributions to mimic how a real radiologist will do the report generation process in a more realistic work environment. All these contributions as listed in section one are valid scientific contributions. I think there should be some credits for that, advancing from the previous literature.

**Weaknesses:**

However I would argue, given the current technical roadmap setup as demonstrated in this paper, you can generate fairly interesting generated reports as results, but this is probably on the wrong technical path to build a automatic reporting system for reliable clinical report generation, no matter you use ChatGPT or even GPT4-V models.

The essential problem/challenges in accurate and automatic clinical report generation are at the core of detecting/localizing all pathological or clinically significant findings, then classifying/diagnosing these findings acurately and finally forming all findings with diagnosis into a report where the physician can verify and modify for the final use (as a real world clinical adoption).

1, you need be abe to extract/balidate fairly accurate finding labels from training reports using NLP tools (maybe chatGPT, maybe special bio-NLP tools).

2, you need to solve the weakly supervised localization/detection issue by mapping the labels to the image regions (which is called visual grounding).

3, Hopefully with powerful and robust NLP and vision tools, you can curate a dataset by integrating/iterating the above two steps.  Then you need to a train vision encoder/decoder to find pathologies with desirably accurate results on new images to generate an initial report.

4, Doing clinical diagnosis on these findings and comparing the current findings to previous studies to derive the temporal change information (you need to build a classifier according ontology for dignosis and image matching/alignment modules by tracking these findings over time).

5, forming a report and providing means (hyperlinks) for human physicians to inspect and accept and edit the report.

The above steps are logically impossible to bypass if you want to build a useful clinical assist tool in the real world. Your paper as currently does not do the above items. I am not convinced if the goal is to build a clinically viable report generation tool, how would you be able to achieve that.

**Questions:**

Please answer the weakness as provided above. Many if not most of report generation papers no matter where they publish do not understand underlying what are clinically essential informations where report should have, and how ....

---

> ### Comment · Reviewer_NXpC · 2023-11-20
> **additional questions:**
>
> 1, in section 4, not sure how ChatGPT can modify the reports with significant factual changes without "looking at" the images and hopefully with a super high quality visual grounding to accept/reject the clinically significant findings. These revisions are for what purpose? how reliable it can be ... will these revision can potentially cause miss-diagnosis?
>
> 2, "previous visit as context": how a random report can serve as pseudo previous report? According the information theory, when there is no new information, just keep the way it is as an empty prior. Injecting a wrong previous resport seems potentially causing diagnosis errors?
>
> 3, using a medical professional to validate a subset of generated data is not really a reliable approach. Reading check x-ray is not an easy task for radiologists and they can have large inter-observer variations.
>
> 4, in Table 2, it is nice and convenient to have all functions. I can see the radiologists could have convenience on editing/producing reports but by reading into the details on these actual report contents, how you guarantee high accuracy in the content it generated/modified. If these reports are for the true purpose of performing diagnosis on real patients, should we trust the critical clinical accuracy of the contents generated from Table 2, or you are just showing the functions?
>
> Last but not least, the overall technical novelty and contributions of this submission is very incremental, too much on using ChatGPT type of tool for such a report generation purpose. This may not be sufficiently competitive for ICLR standard.

---

> ### Author Response · Authors · 2023-11-22
> **Response to Reviewer NXpC (Part 1)**
>
> Thanks for your insightful comments regarding technical roadmap and other valuable questions.
>
> ### Response to technical path
>
> > However I would argue, given the current technical roadmap setup as demonstrated in this paper, you can generate fairly interesting generated reports as results, but this is probably on the wrong technical path to build an automatic reporting system for reliable clinical report generation, no matter you use ChatGPT or even GPT4-V models...The above steps are logically impossible to bypass if you want to build a useful clinical assist tool in the real world. Your paper as currently does not do the above items. I am not convinced if the goal is to build a clinically viable report generation tool, how would you be able to achieve that.
>
>
> - **Benefits in our technical path** We agree that what you listed is a reliable and accurate way of building a report generation pipeline, but it lacks the flexibility of differnt format of input or output. This flexibility of taking different forms of context input and output in different forms for report generation is what we really mean by saying "real-world", and we apologize if this leads to any confusion. The technical paths of training language models end-to-end has the potential and flexibility of doing such tasks. Given the strong language understanding and generation ability, leveraging a pretrained LLM would enable diverse, comprehensive and long input and output with good generalizability. Our proposed approach of adapting multimodal language model is able to customize the model for this flexibility as long as you have the training data, and one of our main contributions is just to create these data from existing text report. We also want to emphasize that our technical path also has good potential for further improving accuracy and robustness and even for visual output in the future.
> - **Existing works mainly follow similar technical path**  Also, many previous works also went this path of using multimodal LLMs for medical VQA and report generation, such as Microsoft's LLaVa-Med [4] and Google's Med PaLM M [5], Stanford's Med-Flamingo [6], therefore we also want to explore in this technical path.
> - **Achieve/bypass the traditional steps of building report generation system** Below we discuss the 5 steps you mentioned for building a report generation system in detail and how this technical path of training end-to-end using text-image paired data achieves the critical functionality or bypass some unnecessary steps.

---

> ### Author Response · Authors · 2023-11-22
> **Response to Reviewer NXpC (Part 2)**
>
> - **Label extraction and visual grounding.** First of all, you mentioned label extraction and visual grounding in the context of creating a dataset that including the localization information that could help generate accurate report. We are not sure why you mention weakness 1 and 2, I guess your belief is that training data should be as accurate and finegrained as pathological labels and corresponding regions, but not a general free-text dataset such as MIMIC-CXR? We believe these two are equivalent, and in many cases image-text pair data is the most and easiest you can get. Given the strong potential of LLM, we believe that these steps can be skipped if your goal is to generate a text report and don't care any visual or structured label output. Most previous works that are language model based cited in our paper do not need data other than image and report text. As an example for location, in our generated report template data, there are some generated templates that require the physician/model to fill in location or indicate absence of each possible pathology label, and our trained model can correctly identify the location most of the time:
>      > TEMPLATE:
>         >- cardiomediastinal contours:
>         >- atelectasis:
>         >- location:
>         >- effusion:
>         >- location:
>         >- interstitial edema:
>         >- location:
>         >- medical devices:
>         >- location:
>         >- type:
>         >
>         > PREDICTED:
>         >- cardiomediastinal contours: stable in appearance.
>         >- atelectasis: worsening predominantly right-sided bibasilar atelectasis.
>         >- location: bibasilar
>         >- effusion: small pleural effusions
>         >- location: right
>         >- interstitial edema: interstitial edema has resolved.
>         >- medical devices:
>         >- location: left internal jugular
>         >- type: central venous catheter.
>         >
>         >GT:
>         >- cardiomediastinal contours: stable appearance
>         >- atelectasis: present
>         >- location: bibasilar
>         >- effusion: present
>         >- location: small right pleural effusion
>         >- interstitial edema: persistent
>         >- medical devices: left internal jugular catheter remains in place within the left superior vena cava.
>
>     Although not the main focus of this paper, we agree that visual grounding is a very important task in medical image analysis, and we hope to enhance our model with that ability in future works. Multimodal LLMs are shown to be able to do visual grounding tasks in general domain [1,2,3], therefore we believe there are ways to adapt this ability to medical domain as well.
>  - **Comparing with previous studies using image alignment moduel.** Second, we believe an image matching/alignment module is also not necessary and can be bypassed. Since our model is based on Flamingo, which is able to receive multiple images or even video as input, we are able to input multiple images and previous reports with chronological order to naturally derive the temporal change information. This is included in our tasks in the paper. Previous non-LLM works cited in our paper and general response also achieved this and did not use any image alignment module.
>   - **Human editing.** Lastly, 5 seems like a deployment system design issue and not in our scope of discussion about report generation, however, we include a task that simulates this clinical procedure in our paper.

---

> ### Author Response · Authors · 2023-11-22
> **Response to Reviewer NXpC (Part 3)**
>
> ### Response to report revision (additional question 1)
>
> > in section 4, not sure how ChatGPT can modify the reports with significant factual changes without "looking at" the images and hopefully with a super high quality visual grounding to accept/reject the clinically significant findings. These revisions are for what purpose? how reliable it can be ... will these revision can potentially cause miss-diagnosis?
>
> Indeed we use ChatGPT to modify the report without looking at images, but we think there is a potential misunderstanding. In this task we only require ChatGPT to generate a modified report and the instruction of how to change it back into the ground truth report as context input. The ground truth report for training is untouched in this task. The purpose of this is to simulate the "real-world" clinical scenario where the physician needs to modify an incorrect report in a comprehensive way, probably also referencing the images (such as the scenario you mentioned in weakness 5). The input is a report that needs to be modified and the corresponding instruction to modify it (these are generated by ChatGPT). The instruction can sometimes be very vague such as "change the location of catheter tube in the report", and therefore the model should be able to reference the image for correct location. These revisions will not cause miss-diagnosis because the paired ground truth report and images are not modified, only the input instruction and context is generated.
>
> ### Response to previous visit as context (additional question 2)
>
> > "previous visit as context": how a random report can serve as pseudo previous report? According the information theory, when there is no new information, just keep the way it is as an empty prior. Injecting a wrong previous resport seems potentially causing diagnosis errors?
>
>  - We do not use random or any generated previous report in our experiment. MIMIC-CXR has the metadata that includes patient identifier and chronological order of the reports, therefore we directly parse the metadata and retrieve the previous report within MIMIC-CXR dataset.
>  - We also propose an approach to generate "pseudo report" when this kind of metadata is not available. For example if the ground truth report have some descriptions like "compared to previous report, the pleural effusion is resolved", then ChatGPT generated pseudo previous report should include diagnosis of pleural effusion. This is not random but based on the description in the current report.
>
> ### Response to data validation (additional question 3)
>
> > using a medical professional to validate a subset of generated data is not really a reliable approach. Reading check x-ray is not an easy task for radiologists and they can have large inter-observer variations.
>
> Please also refer to our general response. We agree that inter-observer variations would be large, but given the unmodified ground truth report and medical images for reference, it would be relatively easy to validate the generated data. Also, we will employ some cross-validation to try to reduce the variation.
>
>
> ### Response to clinical accuracy (additional question 4)
>
> >...how you guarantee high accuracy in the content it generated/modified. If these reports are for the true purpose of performing diagnosis on real patients, should we trust the critical clinical accuracy of the contents generated from Table 2, or you are just showing the functions?
>
> Among the 5 tasks, 4 of them directly use the ground truth from MIMIC-CXR dataset unchanged. The template task uses the generated ground truth, but the modification should be only on the format and the clinical meaning should not have changed by ChatGPT. We have validated on small subset and we will perform further larger-scale validation and cleaning before releasing the data.
>
> ### Response to technical contribution
>
> > Last but not least, the overall technical novelty and contributions of this submission is very incremental...
>
> Please refer to our general response regarding technical novelty and contribution.
>
>
> Reference:
>
> [1] Wang, Wenhai, et al. "Visionllm: Large language model is also an open-ended decoder for vision-centric tasks." arXiv preprint arXiv:2305.11175 (2023).
>
> [2] Zhao, Yang, et al. "Bubogpt: Enabling visual grounding in multi-modal llms." arXiv preprint arXiv:2307.08581 (2023).
>
> [3] Zhou, Qiang, et al. "InfMLLM: A Unified Framework for Visual-Language Tasks." arXiv preprint arXiv:2311.06791 (2023).
>
> [4] Li, Chunyuan, et al. "Llava-med: Training a large language-and-vision assistant for biomedicine in one day." arXiv preprint arXiv:2306.00890 (2023).
>
> [5] Tu, Tao, et al. "Towards generalist biomedical ai." arXiv preprint arXiv:2307.14334 (2023).
>
> [6] Moor, Michael, et al. "Med-flamingo: a multimodal medical few-shot learner." arXiv preprint arXiv:2307.15189 (2023).

---

### Author Response · Authors · 2023-11-22
**General Response to Reviewers (Part 3)**

- **Statistical validation and significance testing** We selected the ChatCAD+ model as the baseline to conduct statistical significance test against ours, across five tasks on multiple metrics. The Precision, Recall and F1 metrics were not included in the comparison because they are composite indicators for multi-classification and are not applicable in this case. BLEU@1, BLEU@2, BLEU@3, BLEU@4, METEOR and ROUGE-L metrics were computed 95% confidence interval and tested. The test sample size was substantial, and the differences were quite noticeable. All these metric differences showed statistical significance, with p-values < 0.001. The detailed comparisons are shown in the following table:

    |    | task       | method   | BLEU@1              | BLEU@2              | BLEU@3              | BLEU@4              | METEOR              | ROUGE_L             |
    |---:|:-----------|:---------|:--------------------|:--------------------|:--------------------|:--------------------|:--------------------|:--------------------|
    |  0 | previous report | ChatCAD+  | 0.241 (0.236,0.245) | 0.134 (0.130,0.137) | 0.076 (0.073,0.079) | 0.042 (0.039,0.045) | 0.304 (0.299,0.308) | 0.179 (0.176,0.182) |
    |  1 | previous report | ours     | 0.298 (0.290,0.306) *p<0.001 | 0.181 (0.174,0.187) *p<0.001 | 0.106 (0.099,0.112) *p<0.001 | 0.064 (0.058,0.070) *p<0.001 | 0.276 (0.269,0.283) *p<0.001 | 0.236 (0.230,0.242) *p<0.001 |
    |  2 | revision | ChatCAD+  | 0.521 (0.512,0.530) | 0.459 (0.450,0.469) | 0.414 (0.405,0.424) | 0.378 (0.368,0.388) | 0.700 (0.692,0.708) | 0.556 (0.547,0.566) |
    |  3 | revision | ours     | 0.891 (0.884,0.898) *p<0.001 | 0.868 (0.860,0.875) *p<0.001 | 0.847 (0.839,0.855) *p<0.001 | 0.828 (0.820,0.837) *p<0.001 | 0.905 (0.898,0.911) *p<0.001 | 0.902 (0.896,0.909) *p<0.001 |
    |  4 | no-context | ChatCAD+  | 0.292 (0.289,0.296) | 0.154 (0.151,0.157) | 0.082 (0.079,0.084) | 0.041 (0.038,0.043) | 0.278 (0.275,0.282) | 0.194 (0.191,0.196) |
    |  5 | no-context | ours     | 0.326 (0.321,0.332) *p<0.001 | 0.197 (0.192,0.202) *p<0.001 | 0.115 (0.110,0.120) *p<0.001 | 0.070 (0.065,0.074) *p<0.001 | 0.290 (0.285,0.295) *p<0.001 | 0.239 (0.234,0.243) *p<0.001 |
    |  6 | medical record    | ChatCAD+  | 0.139 (0.135,0.143) | 0.061 (0.059,0.064) | 0.022 (0.020,0.024) | 0.010 (0.008,0.011) | 0.206 (0.201,0.210) | 0.108 (0.106,0.111) |
    |  7 | medical record    | ours     | 0.302 (0.291,0.313) *p<0.001 | 0.183 (0.174,0.191) *p<0.001 | 0.108 (0.101,0.115) *p<0.001 | 0.065 (0.058,0.071) *p<0.001 | 0.275 (0.266,0.284) *p<0.001 | 0.235 (0.228,0.242) *p<0.001 |
    |  8 | template   | ChatCAD+  | 0.304 (0.296,0.312) | 0.246 (0.238,0.253) | 0.204 (0.197,0.211) | 0.169 (0.162,0.176) | 0.450 (0.441,0.458) | 0.285 (0.277,0.292) |
    |  9 | template   | ours     | 0.621 (0.610,0.632) *p<0.001 | 0.555 (0.543,0.567) *p<0.001 | 0.502 (0.489,0.514) *p<0.001 | 0.455 (0.442,0.468) *p<0.001 | 0.631 (0.620,0.642) *p<0.001 | 0.605 (0.594,0.616) *p<0.001 |

- **Large scale validation** We admit that medical domain data is more sensitive and therefore need careful validation by medical specialists. As we mentioned in section 4.3, we invite medical professionals to validate our generated data and find that most generated data are consistent with the ground truth radiology report without factual errors. Due to the time constraint, we only have a small random sampled subset that undergoes clinical validation. We will have more human validations, including large scale validation through crowdsourcing medical professionals, in the future before we release the data.


Reference:

[1] Zhang, Renrui, et al. "Llama-adapter: Efficient fine-tuning of language models with zero-init attention." arXiv preprint arXiv:2303.16199 (2023).

[2] Bannur, Shruthi, et al. "Learning to exploit temporal structure for biomedical vision-language processing." Proceedings of the IEEE/CVF Conference on Computer Vision and Pattern Recognition. 2023.

[3] Santeramo, Ruggiero, Samuel Withey, and Giovanni Montana. "Longitudinal detection of radiological abnormalities with time-modulated LSTM." Deep Learning in Medical Image Analysis and Multimodal Learning for Clinical Decision Support: 4th International Workshop, DLMIA 2018, and 8th International Workshop, ML-CDS 2018, Held in Conjunction with MICCAI 2018, Granada, Spain, September 20, 2018, Proceedings 4. Springer International Publishing, 2018.

[4] Wang, Yizhong, et al. "Self-instruct: Aligning language model with self generated instructions." arXiv preprint arXiv:2212.10560 (2022).

[5] Liu, Haotian, et al. "Visual instruction tuning." arXiv preprint arXiv:2304.08485 (2023).

[6] Li, Chunyuan, et al. "Llava-med: Training a large language-and-vision assistant for biomedicine in one day." arXiv preprint arXiv:2306.00890 (2023).

---

### Author Response · Authors · 2023-11-22
**General Response to Reviewers (Part 2)**

### Contributions beyond technical novelty
We highlight our major part of contribution in the creation of new tasks and dataset benchmark, as well as the evaluation and analysis of the dataset.
- **Novel problem setting.** To our knowledge, we believe we are the first work to thoroughly exploit the possibility to enable the report generation model with **real-world report generation ability**, which **follows flexible instructions** from radiologists and considers **various types of context** information. There are some previous works that consider patient's previous visit as context [2,3] to improve generation accuracy, but none of the works provide a set of tasks comprehensive enough to cover most real-world report scenarios.
- **New Benchmark dataset.** We provide the first real-world report generation benchmark dataset, and we benchmark several state-of-the-art methods on their real-world report generation capability. We also believe our work is the first to leverage the ability of ChatGPT or similar as a tool for data generation that tries to mimic several real-world report generation scenarios, using simple-structured image-text pair data. Many outstanding works published on top conferences also leverage GPT or similar LLMs for text data generation and claim as a main contribution [4,5,6]. Our data generation pipeline is different from all of them with the focus on real-world report generation tasks data, therefore we believe our contribution is significant.


### Data generated lacks validation

- **Quality analysis** To further assess the quality of generated dataset, this time we sampled 100 more cases of all tasks for a manual validation by a medical professional. We update this result in the appendix of the paper. The data quality for samples:

    | Type          | Number |
    | --------------| ------ |
    | Acceptable    | 74     |
    | Tolerable     | 11     |
    | Unacceptable  | 15     |
    | Total Samples | 100    |

    - Acceptable：
    Out of the samples inspected, 74 were deemed to be of good quality. Both instructions and generated contents were accurate.
    - Tolerable：
    Eleven cases were considered tolerable and carried no factual mistakes. For instance：
        - A case with a positive pollen allergy test in the prediction of medical history and exam results without any indications from the original report.
        - A case generated a template which did not have accurate indentation.
    - Unacceptable：
    Fifteen cases were identified as potentially containing mistakes and were considered unacceptable. Some cases with incorrect generated instructions or contents. A few cases that generated reports not following the provided template.

- **Fully validated test set** Besides validating the generated data, we have validated a small scale test dataset including all the real-world tasks mentioned in our paper with 20 samples for each task, total of 100 samples. Based on the ground truth medical report, we let a specialist validate or manually create report template data and report correction data. We use the MIMIC-CXR ground truth data with previous report, and ground truth reports that have an indication section (reason of examination) to serve as ground truth medical history/condition context data. The results obtained from testing on this validated test set serve as supplementary findings to the test results. We update this result in the paper appendix. On this fully validated test set, our trained model have the following performance:

    |ChatCAD+| B@1 | B@2 | B@3 | B@4 | M   | R-L | P   | R   | F1  |
    |-|-----|-----|-----|-----|-----|-----|-----|-----|-----|
    |No Context|0.312|0.152|0.074|0.043|0.283|0.186|0.385|0.500|0.418|
    |Revision|0.828|0.779|0.730|0.690|0.838|0.803|0.721|0.936|0.814|
    |Template|0.274|0.218|0.178|0.143|0.489|0.270|0.471|0.825|0.600|
    |Previous Report|0.302|0.174|0.113|0.079|0.334|0.215|0.438|0.471|0.454|
    |Medical Record|0.166|0.094|0.055|0.035|0.028|0.151|0.282|0.511|0.364|

    |Ours| B@1 | B@2 | B@3 | B@4 | M   | R-L | P   | R   | F1  |
    |-|-----|-----|-----|-----|-----|-----|-----|-----|-----|
    |No Context|0.404|0.257|0.178|0.133|0.327|0.278|0.475|0.380|0.422|
    |Revision|0.929|0.913|0.898|0.883|0.931|0.939|0.978|0.957|0.967|
    |Template|0.634|0.543|0.471|0.412|0.637|0.565|0.892|0.805|0.846|
    Previous Report|0.343|0.213|0.143|0.107|0.287|0.250|0.500|0.449|0.478|
    |Medical Record|0.377|0.218|0.134|0.087|0.299|0.245|0.459|0.395|0.425|

    The result shows no significant deviation from our reported result on generated test set and outperforms the ChatCAD+ baseline on F1 score as well, which shows that our model trained on generated data performs well on real validated test set as well, which indirectly proves that the gap between generated data and real validated data is small.

---

### Author Response · Authors · 2023-11-22
**General Response to Reviewers (Part 1)**

Thanks for all the valuable and constructive feedback from reviewers. We really appreciate your time and effort. Here we would like to first address some common concerns:

### The technical novelty and contribution is incremental

We believe DeMMo is technically novel in terms of both the technical challenges it tackles, and its fundamental design.
- **Technical challenges**. Finetuning an existing LLM to have visual understanding ability in medical domain poses two significant challenges that can not be trivially solved.
  1. **High training cost**: Directly finetuning a Multimodal LLM is surely not an effective way since the number of parameters is more than 8 billion. Hence we propose a parameter-efficient finetuning method to adapt general Multimodal LLM to medical domain by incorporating a medical domain visual encoder.
  2. **Balance between generalization ability and domain-specific capability**: Finetuning the multimodal on small amount of medical data causes model to overfit and lose generalization ability. Either trivially replacing the original encoder with a medical domain encoder or adding another set of adapter (in our case Flamingo perceiver resampler) would not give sufficient performance, as shown in Table 5 and 6 in our paper.


- **Novelty in our design**
  1) **Domain-specific encoder grafting**: We are the first to propose a framework that grafts an additional domain-specific visual encoder onto the pretrained general multi-modal LLM by efficiently tuning adaptors and prompts.  We believe this framework is insightful for other future works that may want to explore a similar technical path of employing pretrained domain-specific modules on multimodal LLMs.
  2) **Patchwise domain-specific prompt tuning**: Instead of one global feature in most existing multi-modal LLM, we use grid features to enhance the model's attention towards critical regions in the medical images. Moreover, specific prompt tokens are learned to be associated with a specific image patch. The whole adaptive prompt is also for the model to learn to distinguish between medical and general domain encoder features.

---

### Meta-Review · Area_Chair_tHbK · 2023-12-02

**Metareview:**

This submission receives the following scores: 3, 5, 5, 8, 5. And the reviewer who gave the score 8 has a low confidence (score 2), whereas other reviewers who incline to reject this submission have a higher confidence (such as score 4 or 5).

The manuscript introduces the Domain-enhanced Multi-modal Model (DeMMo), an approach to radiology report generation tailored for real-world clinical settings. DeMMo integrates a medical domain vision encoder with a general domain multimodal large language model (LLM). This integration aims to utilize the specific capabilities of the medical domain encoder alongside the versatility of the general LLM, enhancing the accuracy and relevance of the generated reports. The model's design reflects the complexity of real-world scenarios, where radiology report generation involves not just image processing, but also adherence to radiologist instructions and contextual data integration, a shift from traditional report generation models.

To support this approach, the authors developed the MIMIC-R3G dataset using a GPT-based data generation pipeline, filling a gap in existing datasets that lack instructional and contextual data for real-world applications. DeMMo is built upon the Flamingo model, with a domain-specific medical encoder enhancing its ability to interpret medical imagery. The model's performance is assessed using the MIMIC-R3G dataset, focusing on clinically relevant findings. DeMMo's effectiveness is measured using natural language generation metrics and clinical efficacy metrics, where it demonstrates competitive performance against existing report generation models. This advancement represents a fair contribution to the field of computer-aided diagnosis systems.

Overall,  the technical contribution is, albeit interesting, considered to be incremental. Furthermore, there is a strong concern about the clinical translatability (see "Why Not Higher Score" below).

**Justification For Why Not Higher Score:**

- Incremental Technical Contribution: Reviewers critiqued the technical novelty of DeMMo as incremental, suggesting it may not significantly advance the existing state-of-the-art in automatic radiology report generation.
- Clinical Translatability Concerns: There were doubts about the model's effectiveness in clinical settings. Reviewers argued that DeMMo, like other similar models, might not be clinically useful due to its reliance on generated data, which might not capture the complexity and nuances of real-world medical data. This is a strong concern.
- Data Generation and Validation Issues: The model's training and validation on generated data raised questions about its ability to handle real, varied clinical scenarios. The limited validation by medical professionals, as noted in the manuscript, was seen as insufficient for establishing the model's clinical accuracy.

**Justification For Why Not Lower Score:**

- Advancement in Real-World Report Generation: Despite critiques, DeMMo represents a good step in addressing real-world clinical reporting needs, a notable gap in previous literature.
- Competitive Performance: The model demonstrated superior performance in clinical efficacy metrics compared to existing models, indicating its potential utility in practical applications.
- Interesting Approach: The integration of a medical domain-specific encoder into a general domain LLM, and the creation of a comprehensive, real-world-focused dataset (MIMIC-R3G), are viewed as a fair contribution to the field.

---

### Decision · Program_Chairs · 2024-01-16

Reject